# GenVP: Generating Visual Puzzles with Contrastive Hierarchical VAEs

**Kalliopi Basioti[1], {Pritish Sahu[2]\*, Qingze Tony Liu[1]}, Zihao Xu[1], Hao Wang[1],
Vladimir Pavlovic[1]**
[1]Rutgers University, [2]SRI International
{kalliopi.basioti,tony.liu,zihao.xu,vladimir}@rutgers.edu
pritish.sahu@sri.com, hw488@cs.rutgers.edu

## Abstract

Raven's Progressive Matrices (RPMs) is an established benchmark to examine the ability to perform high-level abstract visual reasoning (AVR). Despite the current success of algorithms that solve this task, humans can generalize beyond a given puzzle and create new puzzles given a set of rules, whereas machines remain locked in solving a fixed puzzle from a curated choice list. We propose Generative Visual Puzzles (GenVP), a framework to model the entire RPM generation process, a substantially more challenging task. Our model's capability spans from generating multiple solutions for one specific problem prompt to creating complete new puzzles out of the desired set of rules. Experiments on five different datasets indicate that GenVP achieves state-of-the-art (SOTA) performance both in puzzle-solving accuracy and out-of-distribution (OOD) generalization in 22 OOD scenarios. Compared to SOTA generative approaches, which struggle to solve RPMs when the feasible solution space increases, GenVP efficiently generalizes to these challenging setups. Moreover, our model demonstrates the ability to produce a wide range of complete RPMs given a set of abstract rules by effectively capturing the relationships between abstract rules and visual object properties.

## 1 Introduction

Human reasoning involves decision-making, concluding and learning from experiences, knowledge, and sensory input such as visual cues. Various intelligence tests, notably Raven Progressive Matrices (RPM) (Raven & Court, 1998), assess this capability. Originally for humans, RPMs now also test AI's abstract visual reasoning capabilities. Standard RPM puzzles include two parts, a context matrix, and a choice list. The context matrix presents a problem prompt for an incomplete $3 \times 3$ puzzle with a missing bottom-right image, and the choice list provides eight potential solutions.

Recent developments in learning-based RPM solvers comprised two categories: discriminative and generative. Discriminative approaches discern the correct answer among the list of choices using visual features extracted from the RPM puzzle (Zhang et al., 2019a;b; Wu et al., 2020; Benny et al., 2021; Hu et al., 2021; Sahu et al., 2022b; Małkiński & Mańdziuk, 2022; Yang et al., 2023; Xu et al., 2023; Mondal et al., 2023; Yang et al., 2023; Zhao et al., 2024). Although high in accuracy, discriminative solvers are limited by the choice list, fixed in size and limited in search space, often leading to short-cut learning (Zhuo & Kankanhalli, 2020; Zhang et al., 2019b). Generative approaches (Pekar et al., 2020; Sahu et al., 2022a) alleviate these issues by generating a proposed solution and retrieving the closest matching element from the choice list. Despite good performance and desirable properties, generative approaches (Pekar et al., 2020; Zhang et al., 2021; Sahu et al., 2022a; Shi et al., 2024) are sensitive to noisy and distractive puzzle attributes, lacking the ability to distinguish relevant image features. Furthermore, these solvers are restricted to image-level generation, demonstrating a limited high-level abstraction capability.

We propose Generative Visual Puzzles (GenVP), a generative RPM approach that models the entire RPM generation process. GenVP achieves this through a graphical model that contains a hierarchical inference and generative pipeline which are trained simultaneously. GenVP graphical structure

---

\*Authors named inside {} equally contributed.

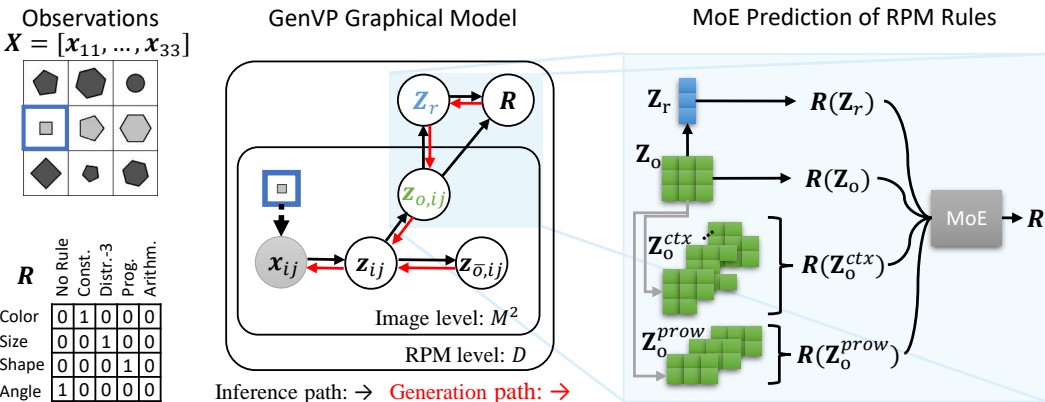

Figure 1: Generative and Inference Graphical Models of GenVP. During inference, we are given a complete, valid puzzle $\mathbf{X}$ and our task is to infer all intermediate random variables $\mathbf{Z}, \mathbf{Z}_o, \mathbf{Z}_{\bar{o}}, \mathbf{Z}_r$ and predict its rules $\mathbf{R}$ using our Mixture of Experts (MoE) strategy. During generation, given an abstract set of rules $\mathbf{R}$, we generate a complete puzzle $\mathbf{X}$ following the desired rules. We note that during generation, we don't need the MoE module, but we directly generate $\mathbf{Z}_r$ from the given set of rules.

introduces a powerful Mixture of Experts (MoE) mechanism for puzzle rule prediction, improving the model's resilience to noisy features and shortcut learning. Our approach can also generate complete RPM matrices, demonstrating a deep understanding of the puzzle rules. We robustify our generative model training by designing a novel contrastive learning scheme, leading to cross-puzzle and cross-candidate comparisons. In summary, our contributions are:

- We propose GenVP, a novel approach for solving and creating visual puzzles. GenVP is the first AVR model to learn a generative path from abstract rules to complete visual puzzles. Our model promotes AI creativity by producing new, complete puzzles, which also serve as high-level, human-readable visual explanations of the learned concepts. This is because we can assess the model's understanding of a rule by prompting it to create a complete puzzle based on that rule.
- We propose a novel cross-puzzle and cross-candidate contrastive loss for AVR. Additionally, we introduce a MoE mechanism to learn robust representations that can accurately deduce the rules of visual puzzles and subsequently solve RPMs.
- Our experiments on five different AVR datasets show that GenVP outperforms state-of-the-art (SOTA) RPM solvers. Furthermore, GenVP significantly surpasses SOTA in 22 out of our 24 examined out-of-distribution (OOD) evaluations.
- Our qualitative analysis indicates that GenVP exhibits a high-level capability for rule abstraction by generating new RPMs that differ from those in the original datasets while following the same set of rules.

## 2 RELATED WORK

**Generative approaches.** GCA (Pekar et al., 2020) pioneered a generative approach for RPM solving, integrating a VAE-based (Doersch, 2016) image reconstructor, used for feature extraction, with a discriminative RPM classifier. It was trained using image reconstruction, rule classification, and contrastive learning for the latent representations of the choice list images; however, the model struggles with low puzzle solution accuracy. DAREN (Sahu et al., 2022a) improved the accuracy by adopting separate training for generative and discriminative components by employing a pre-trained VAE to extract robust image features for the solver module. While DAREN has shown high accuracy, it is limited to the constant attribute rule, making it unsuitable for more complex rules such as arithmetic, progression, and distribute-three.

Later approaches (Zhang et al., 2021; 2022; Shi et al., 2024) aim to generate the missing puzzle image in RPMs. Specifically, PRAE (Zhang et al., 2022) and ALANS (Zhang et al., 2021) employ a neural perception module to produce probabilistic scene representation alongside a symbolic log-

ical reasoning component. These methods improve the performance of RPM solving using strong inductive biases, exploiting specific configuration layouts, or hidden relations within RPMs in their model design. However, the high inductive biases limit generalization capability to novel puzzle setups, such as different layouts, and the models have high computational overhead and scalability issues within their architecture. RAISE (Shi et al., 2024) also generates the missing image but differs from previous works in that it employs special rule–executors to generate images adhering to specific rules predicted from the context matrix. Each non-random rule (constant, progression, arithmetic, and distribute-3) has its dedicated executor module that generates latent features for final image decoding. Despite the SOTA RPM solving accuracy, RAISE suffers from the following drawbacks: a) error propagation: RAISE will use a faulty rule executor when the rule prediction is incorrect; b) RAISE struggles to adapt in puzzle regimes with a large number of feasible solutions, leading to poor puzzle-solving performance in noisy, large solution space puzzles. As the solution space expands, RAISE fails to capture it, leading to an inability to even distinguish the correct solution from the incorrect ones in the choice list. Our experiments also show that RAISE suffers from training instability and poor OOD generalization.

Compared to prior SOTA generative approaches, GenVP learns to focus on relevant factors for the rules and minimizes the impact of distracting attributes on rule prediction. Further, our model employs a general rather than a dedicated module for image generation. Existing approaches can only reconstruct the missing panels of the context matrix, whereas GenVP has broader generative capabilities, including generating multiple solutions for a given puzzle and creating novel complete puzzles from just a set of abstract rules. GenVP learns robust representations by proposing a novel contrasting scheme for the rules of valid and invalid puzzles (the latter are formed by completing the context matrix with incorrect candidates). Moreover, our approach utilizes a robust cluster of rule predictors (MoE) by 1) exploiting various combinations of puzzle images and 2) employing low- and high-level latent variables. Note that each GenVP predictor has to predict all present rules in a puzzle compared. Whereas, in RAISE, the latent space is broken down into concept-specific dimensions, where single rule predictors are applied. Therefore, our MoE does not use strong inductive biases (i.e., prior knowledge of the number of attributes), while at the same time, it increases our model robustness by using partially observed information (a subset of puzzle images) and coarse to fine representations. Finally, GenVP exceeds SOTA generative approaches in puzzle-solving accuracy for both in-distribution and in 22 out of 24 examined OOD cases. The OOD scenarios include novel attribute values, rules, domain transfer and compositional generalization.

**Contrastive Learning in RPMs.** CoPINet (Zhang et al., 2019b), DCNet (Zhuo & Kankanhalli, 2020) and MLCL (Małkiński & Mańdziuk, 2022) were the first to adopt contrastive learning (Chen et al., 2020; Liu et al., 2021) for discriminative RPM solvers. CoPINet and DCNet use contrastive learning for the choice list images but suffer from short-cut learning issues. MLCL utilizes SimCLR (Chen et al., 2020) contrastive loss by creating positive and negative batches through examination of RPM rules (if two puzzles share common rules, they are marked as positive or otherwise negative). However, MLCL uses contrasting only as a separate pre-training step, where the main training relies on a standard cross-entropy objective. In our work, we create novel contrasting terms for rule representations to accompany the generative learning of GenVP. We adopt dual-level contrasting, comparing both global (cross-puzzle) and local (cross-candidate) RPMs, which leads to robust rule estimation.

## 3 METHOD

**Problem Setting.** The Raven Progression Matrix (RPM) consists of $M \times N$ images (usually $M = N = 3$) denoted as $\mathbf{X} = \{\mathbf{x}_{ij}\}_{i \in [M], j \in [N]}$, where each $\mathbf{x}_{ij}$ has dimensions $H$ and $W$. The RPM adheres to a set of rules encoded in $\mathbf{R} \in \{0, 1\}^{K_R \times N_R}$, with rows representing $K_R$ attributes and columns representing one-hot rule encodings. The RPM puzzle includes a context matrix, derived by masking the image $\mathbf{x}_{MN}$ and a choice list containing this masked image plus the negative examples set $\mathbf{A} = \{\mathbf{a}_i | \mathbf{a}_i \in \mathbb{R}^{H \times W}\}_{i=1}^{A}$, where $A$ is the number of negative images (typically $A = 7$ in RPM datasets). The objective is to identify the correct $\mathbf{x}_{MN}$ in the choice list to complete the context matrix.

## 3.1 GENVP LATENT VARIABLES

**Latent Variable Notations.** GenVP includes image-level and RPM/puzzle-level latent variables. For image-level, $\mathbf{Z} = \{\mathbf{z}_{ij}\}_{i\in[M],j\in[N]}$ represents the compact latent representation of RPM images $\mathbf{X}$. $\mathbf{Z}$ is split into $\mathbf{Z}_o = \{\mathbf{z}_{o,ij}\}_{i\in[M],j\in[N]}, \mathbf{z}_{o,ij} \in \mathbb{R}^{K_{Z_o}}$ for relevant RPM attributes, and $\mathbf{Z}_{\bar{o}} = \{\mathbf{z}_{\bar{o},ij}\}_{i,j=1}^M, \mathbf{z}_{\bar{o},ij} \in \mathbb{R}^{K_{Z_{\bar{o}}}}$ for irrelevant attributes, as shown in Fig. 1 (left). Due to insufficient knowledge about full puzzle at the image level, puzzle-level variables $\mathbf{Z}_r$ and $\mathbf{R}$ are introduced to bridge this gap and aid in decoupling of $\mathbf{Z}$, with $\mathbf{R}$ denoting RPM rules and $\mathbf{Z}_r$ capturing the relationships among the $M$ images per row $r$, as depicted in Fig. 1 (middle).

## 3.2 GENERATIVE AND INFERENCE MODELING IN GENVP

In this section, we outline the generative and inference pipeline of GenVP, using previously defined variables. We denote categorical distributions by $\mathcal{C}(\cdot)$, Gaussian distributions by $\mathcal{N}(\cdot)$, and the set index by $[n] = \{1, 2, \ldots, n | n \in \mathbb{N}\}$.

### 3.2.1 INFERENCE MODULE

GenVP inference module is divided into two steps, image-level and RPM-level inference. For the simplicity of notation, we denote all learnable parameters for the inference pipeline as $\phi$.

**Image Level Inference.** Image level inference processes individual images from the context matrix to infer variables $\mathbf{Z}, \mathbf{Z}_o, \mathbf{Z}_{\bar{o}}$ as shown in Fig. 1 (middle). The posterior for these variables follow conditional Gaussian distribution. The means and variances are parameterized by neural networks with learnable parameter $\phi$, and the functional forms are detailed in the Appendix A.

**RPM-Level Inference.** The RPM level representations consist of the rule matrix $\mathbf{R}$ and the rows representation $\mathbf{Z}_r$. We first infer $\mathbf{Z}_r \sim \mathcal{N}\left(\mu_{\mathbf{Z}_r,q}(\mathbf{Z}_o;\phi), \sigma^2_{\mathbf{Z}_r,q}(\mathbf{Z}_o;\phi)\right)$ conditioned on previously inferred $\mathbf{Z}_o$. Using $\mathbf{Z}_r$, we proceed to predict the rules $\mathbf{R}$ for RPM $\mathbf{X}$. We employ an ensemble of predictors to establish a robust rule estimator, assuming that $\mathbf{X}$'s rules $\mathbf{R}$ are derivable from the following combinations of latent variables:

- The $MN$ image latent representations $\mathbf{Z}_o = \{\mathbf{z}_{o,ij}\}_{i\in[M],j\in[N]}$. With posterior

$$q_\phi(\mathbf{R}|\mathbf{Z}_o) = \mathcal{C}\left(\mathbf{R}; p_{\mathbf{R}_1,q}(\mathbf{Z}_o;\phi)\right) \tag{1}$$

- Any $MN - 1$ context latent representations $\mathbf{Z}_o^{ctx} = \mathbf{Z}_o \setminus \mathbf{z}_{o,kl}$ where $k \in [M], l \in [N]$ are the missing image coordinates. With posterior

$$q_\phi(\mathbf{R}|\mathbf{Z}_o^{ctx}) = \mathcal{C}\left(\mathbf{R}; p_{\mathbf{R}_2,q}(\mathbf{Z}_o^{ctx};\phi)\right) \tag{2}$$

- Any $MN - N$ latent representations $\mathbf{Z}_o^{prow} = \mathbf{Z}_o \setminus \mathbf{z}_{o,k:}$, with $k \in [M]$ the row-index of the excluded row (partial rows). With posterior

$$q_\phi(\mathbf{R}|\mathbf{Z}_o^{prow}) = \mathcal{C}\left(\mathbf{R}; p_{\mathbf{R}_3,q}(\mathbf{Z}_o^{prow};\phi)\right) \tag{3}$$

- The $M$ image row latent representations $\mathbf{Z}_r = \{\mathbf{z}_{r,i}\}_{i=1}^M$. With posterior

$$q_\phi(\mathbf{R}|\mathbf{Z}_r) = \mathcal{C}\left(\mathbf{R}; p_{\mathbf{R}_4,q}(\mathbf{Z}_r;\phi)\right). \tag{4}$$

Our final puzzle rules estimation results from a mixture of the aforementioned experts (MoE):

$$q_\phi(\mathbf{R}|\mathbf{Z}_o, \mathbf{Z}_r) = \mathcal{C}\left(\mathbf{R}; \mathbf{p}_{\text{MoE},q}(\mathbf{Z}_o, \mathbf{Z}_r;\phi)\right). \tag{5}$$

For the MoE estimator shown in Fig. 1 (right), we investigated parametric (neural network approximations) and non-parametric solutions. In Appendix C.6, we provide an ablation study for different mixture functions using parametric (neural networks) and non-parametric approaches. The MoE module also ensures that $\mathbf{Z}_o$ and $\mathbf{Z}_r$ learn to capture essential information for rule prediction during training. The model is tasked with identifying the most probable rule set without full visibility of the $M \times N$ puzzle. To enhance the stability of contrastive learning (Section 3.3), separate NN approximators are employed for each predictor in Eqs. (1) to (4).

**RPM Solution Selection.** We observed that the most likely RPM puzzle solution, from a choice of $A + 1$ images (A negatives and one correct), is the one that forms the most rules with the context

matrix. Using this observation, we select the RPM answer by first creating eight complete candidate puzzles, filling the bottom-right empty spot with choice list images. For each candidate we use GenVP inference pipeline to find their MoE rule matrix prediction, then we choose the one having the largest set of active rules (constant, progression, distribute-three, arithmetic) and return it as the final answer.

### 3.2.2 GENERATIVE MODULE

The generative pipeline inputs a rule matrix $\mathbf{R}$ and irrelevant attributes $\mathbf{Z}_{\bar{o}}$, producing complete RPM instances as shown in Fig. 1(middle) with red solid edges. The puzzle rule variable $p(\mathbf{R}) = \prod_k \mathcal{C}(\mathbf{r}_k)$ comprises independent variables $\mathbf{r}_k$ for each attribute $k$, each following a categorical distribution $\mathcal{C}(\mathbf{r}_k)$. Irrelevant attributes $\mathbf{Z}_{\bar{o}}$ are sampled from a standard Gaussian distribution $\mathcal{N}(\mathbf{0}, I)$. The remaining random variables follow conditional multivariate Gaussian distributions, whose conditional relations are illustrated in Fig. 1 (middle), with means and covariances approximated by neural network decoders parameterized by $\boldsymbol{\theta}$, with the functional definitions detailed in the Appendix A.

### 3.3 THE OBJECTIVE FUNCTION

Our GenVP is trained with the combination of a standard Hierarchical VAEs (HVAE) optimization by maximizing the evidence lower bound (ELBO) (Doersch, 2016), and a contrastive learning between different RPM matrices composed of the set of provided negative answers $\mathbf{A}$.

**ELBO.** To learn the encoders and decoders model parameters, we maximize the ELBO of GenVP:

$$\text{ELBO}_{\boldsymbol{\theta},\phi}(\mathbf{X}, \mathbf{R}) = \underbrace{\beta_1 \mathbb{E}_q \left[\log p_{\boldsymbol{\theta}}(\mathbf{X}|\mathbf{Z})\right]}_{\mathcal{L}_{\text{rec}}} + \underbrace{\beta_2 \mathbb{E}_q \left[\log \frac{p(\mathbf{R})}{q_\phi(\mathbf{R}|\mathbf{Z}_o, \mathbf{Z}_r)}\right]}_{\mathcal{L}_R} + \underbrace{\beta_3 \mathbb{E}_q \left[\log \frac{p(\mathbf{Z}_{\bar{o}})}{q_\phi(\mathbf{Z}_{\bar{o}}|\mathbf{Z})}\right]}_{\mathcal{L}_{Z_{\bar{o}}}} +$$
$$\underbrace{\beta_4 \mathbb{E}_q \left[\log \frac{p_{\boldsymbol{\theta}}(\mathbf{Z}_o|\mathbf{Z}_r)}{q_\phi(\mathbf{Z}_o|\mathbf{Z})}\right]}_{\mathcal{L}_{Z_o}} + \underbrace{\beta_5 \mathbb{E}_q \left[\log \frac{p_{\boldsymbol{\theta}}(\mathbf{Z}_r|\mathbf{R})}{q_\phi(\mathbf{Z}_r|\mathbf{Z}_o)}\right]}_{\mathcal{L}_{Z_r}} + \underbrace{\beta_6 \mathbb{E}_q \left[\log \frac{p_{\boldsymbol{\theta}}(\mathbf{Z}|\mathbf{Z}_o, \mathbf{Z}_{\bar{o}})}{q_\phi(\mathbf{Z}|\mathbf{X})}\right]}_{\mathcal{L}_Z}. \quad (6)$$

We list the description for each individual term as follows:

- $\mathcal{L}_{\text{rec}}$ trains the reconstruction of image $X$ using image embedding $\mathbf{z}_{ij} \in \mathbf{Z}$.
- $\mathcal{L}_R$ trains the rule-relevant latent representations, $\mathbf{Z}_o$ and $\mathbf{Z}_r$, to decipher the puzzle rules.
- $\mathcal{L}_{Z_{\bar{o}}}$ ensures the posterior distribution $Z_{\bar{o}}$ follows standard normal distribution.
- $\mathcal{L}_{Z_o}, \mathcal{L}_{Z_r}, \mathcal{L}_Z$ regularize the latent representation and encourage learning of informative latent representation for the three variables.

When rule annotations are accessible during training, then the prior $p(\mathbf{R})$ becomes data-dependent. In particular, for a pair of complete puzzle and rule observations $(\mathbf{X}_i, \mathbf{R}_i)$ we will use a prior $p(\mathbf{R}|\mathbf{R}_i)$ that is highly concentrated around $\mathbf{R}_i$ by approaching a Dirac Delta distribution $p(\mathbf{R}|\mathbf{R}_i) = \delta(\mathbf{R} - \mathbf{R}_i)$. For all loss terms, we use the Kullback-Leibler (KL) divergence (Joyce, 2011) closed form for Gaussian variables, except for the $\mathcal{L}_{\text{rec}}$ and $\mathcal{L}_R$, which reduce to reconstruction losses between the puzzle and rule predictions and their corresponding annotations.

**Contrastive Learning Overview.** We design contrastive terms using the property of boundedness of $\mathbf{R}$ on the $[0, 1]$ interval, with all row sums equal to one. Since the rule variable $\mathbf{R}$ is a leaf node in our graphical model, the contrastive gradients are able to backpropagate and influence all latent variables of GenVP. Our contrastive loss contains two terms, a global masked contrastive loss and a local contrastive loss, which we discuss next.

**Global Masked Contrastive Loss.** We design a global contrastive loss with the aim of learning accurate rule estimations by comparing globally across different puzzles. For simplicity, we contrast two puzzles $\mathbf{P}_1 = (\mathbf{X}_1, \mathbf{R}_1)$ and $\mathbf{P}_2 = (\mathbf{X}_2, \mathbf{R}_2)$ at a time, knowing that the following holds:

1. $\mathbf{R}_1 = \mathbf{R}_2$: puzzles follow the same rules.
2. $\mathbf{R}_1 \neq \mathbf{R}_2$, $\mathbf{R}_{1,i:} \neq \mathbf{R}_{2,i:}$ rows $i, \forall i = 1, \dots K_R$: puzzles follow entirely different rules.

3. $\mathbf{R}_1 \neq \mathbf{R}_2$: puzzles share the same rule for some attributes $|C| < K_R$.

Consequently, we choose to contrast pairs of puzzles from category 2 and to contrast pairs from category 3 only for attributes that have different rules. To achieve this, we design the following masked contrastive loss between pairs of RPM puzzles:

$$g_{\boldsymbol{\theta}}(\mathbf{P}_1, \mathbf{P}_2) = ||\mathbf{M} \odot (\hat{\mathbf{R}}_{\boldsymbol{\theta}}(\mathbf{X}_1) - \mathbf{R}_2)||^2 - ||\bar{\mathbf{M}} \odot (\hat{\mathbf{R}}_{\boldsymbol{\theta}}(\mathbf{X}_1) - \mathbf{R}_2)||^2, \mathbf{M}_{ij} = \begin{cases} 1 \text{ if } \mathbf{R}_{1,ij} \neq \mathbf{R}_{2,ij} \\ 0 \text{ otherwise} \end{cases} \tag{7}$$

where $\hat{\mathbf{R}}_{\boldsymbol{\theta}}(\mathbf{X}_1)$ are the predictions of the rule matrix of the puzzle $\mathbf{X}_1$ using the encoders from Eqs. (1) to (4). We use $\odot$ for element-wise multiplication by masks $\mathbf{M}$ and $\bar{\mathbf{M}}$. The mask $\bar{\mathbf{M}}$ is designed to have the opposite effect of $\mathbf{M}$; it is equal to one when the two rows follow the same rules and zero otherwise.

$$\bar{\mathbf{M}}_{ij} = \begin{cases} 1, \text{ if } \mathbf{R}_{1,ij} = \mathbf{R}_{2,ij} \text{ and } \mathbf{R}_{2,ij} = 1 \\ 0, \text{ otherwise.} \end{cases} \tag{8}$$

We can generalize Eq. (7) to a batch with size $B$ for each puzzle $\mathbf{X}_b, b \in [B]$ by

$$\mathbf{G}_{\boldsymbol{\theta}}(\mathbf{P}_1, \mathbf{P}_2, \ldots \mathbf{P}_B) = \frac{1}{B} \sum_{b \in [B]} \left( \frac{1}{B-1} \sum_{i \in [B], i \neq b} g_{\boldsymbol{\theta}}(\mathbf{P}_b, \mathbf{P}_i) \right). \tag{9}$$

**Local Masked Contrastive Loss.** ELBO objective and global contrastive loss promote learning from correct puzzles. However, these objectives lack contrast with invalid/incorrect ones. To make the model aware of the invalid puzzles, we introduce a local masked contrastive loss between the valid puzzle $\mathbf{X}$ and a set of invalid puzzles $\mathbf{X}^- = \{\mathbf{X}_i^-\}_{i=1}^A$, obtained by replacing masked $\mathbf{x}_{MN}$ with each negative image $\mathbf{a}_i$ from the choice list. Since the choice list is generated by modified rule-relevant attributes, we are able to adjust the rule matrix $\mathbf{R}$ to $\mathbf{R}_i^-$ by setting the rule value for modified attributes to 'random/no rule'. We then form a tuple of negative puzzle and perturbed rule matrix $\mathbf{P}_i^- = (\mathbf{X}_i^-, \mathbf{R}_i^-)$ and establish the local masked contrastive loss between a valid puzzle $\mathbf{P}$ and an invalid one $\mathbf{P}_i^-$ as follows:

$$\mathbf{L}_{\boldsymbol{\theta}}(\mathbf{P}, \mathbf{P}_1^-, \ldots \mathbf{P}_A^-) = \sum_{i \in [A]} g_{\boldsymbol{\theta}}(\mathbf{P}_i^-, \mathbf{P}). \tag{10}$$

**Training.** During training, we start with a 'warm-up' phase that maximizes only ELBO, Eq. (6). Afterward, we continue training with ELBO and the global-local masked contrastive terms:

$$\max_{\boldsymbol{\theta}, \boldsymbol{\phi}} \left( \frac{1}{B} \sum_b \left( \text{ELBO}_{\boldsymbol{\theta}, \boldsymbol{\phi}}(\mathbf{X}_b, \mathbf{R}_b) - \mathcal{L}_{sup} + \frac{\beta_G}{B-1} \sum_{i \in B/b} g_{\boldsymbol{\theta}}(\mathbf{P}_b, \mathbf{P}_i) + \beta_L \mathbf{L}_{\boldsymbol{\theta}}(\mathbf{P}_b, \mathbf{P}_1^-, \ldots \mathbf{P}_A^-) \right) \right) \tag{11}$$

where $\beta_G, \beta_L$ are scalar hyper-parameters tuning the effect of the contrastive terms.

Data augmentation has been shown to play a crucial role in the success of contrastive learning (Chen et al., 2020; Liu et al., 2021). For this reason, we adapted rule-invariant puzzle augmentations from Zhang et al. (2019b); Małkiński & Mańdziuk (2022); Zhao et al. (2024) during training. Types of such augmentations include performing image transformations like horizontal or vertical flips, rolling, and grid shuffling.

## 4 EXPERIMENTAL SETUP

**Data.** We assessed GenVP with the RAVEN-based (RAVEN (Zhang et al., 2019a), I-RAVEN (Hu et al., 2021) and RAVEN-FAIR (Benny et al., 2021)) and the VAD (Hill et al., 2019) and PGM (Barrett et al., 2018). Developing efficient generative approaches for the last two datasets poses a significant challenge due to the vast number of feasible solutions for each puzzle (Shi et al., 2024). For more dataset details see Appendix B.

Table 1: RPM solving accuracy (%) for RAVEN/I-RAVEN datasets. *RAISE*: reproduced results*

| MODEL | AVG | C-S | L-R | U-D | O-IC | O-IG | 2×2 | 3×3 |
|---|---|---|---|---|---|---|---|---|
| GCA | 32.7/41.7 | 37.3/51.8 | 26.4/44.6 | 21.5/42.6 | 30.2/46.7 | 33.0/35.6 | 37.6/38.1 | 43.0/32.4 |
| ALANS | 54.3/62.8 | 42.7/63.9 | 42.4/60.9 | 46.2/65.6 | 49.5/64.8 | 53.6/52.0 | 70.5/66.4 | 75.1/65.7 |
| PRAE | 80.0/85.7 | 97.3/99.9 | 96.2/97.9 | 96.7/97.7 | 95.8/98.4 | 68.6/76.5 | 82.0/84.5 | 23.2/45.1 |
| RAISE | 90.0/92.1 | 99.2/99.8 | 98.5/99.6 | 99.3/99.9 | 97.6/99.6 | 89.3/96.0 | 68.2/71.3 | 77.7/78.7 |
| RAISE* | 89.0/92.3 | 98.7/100 | 99.1/99.2 | 97.7/99.0 | 94.5/97.7 | 79.1/85.6 | 73.4/83.5 | 79.4/82.3 |
| GENVP | 94.7/96.1 | 100/100 | 99.7/99.8 | 99.9/100 | 99.9/100 | 86.0/90.1 | 93.3/94.6 | 84.1/88.1 |

Table 2: OOD RPM solving accuracy (%) for puzzles with unseen attribute values. -I indicated interpolating values and -E for extrapolating values. IN-DIST is the accuracy for the original RAVEN/I-RAVEN testing sets (without unseen attributes). *RAISE*: reproduced results, original paper did include this evaluation.*

| OOD | MODEL | AVG | C-S | L-R | U-D | O-IC | O-IG | 2×2 | 3×3 |
|---|---|---|---|---|---|---|---|---|---|
| IN-DIST | RAISE* | 89.0/92.3 | 98.7/100 | 99.1/99.2 | 97.7/99.0 | 94.5/97.7 | 79.1/85.6 | 73.4/83.5 | 79.4/82.3 |
| | GENVP | 94.7/96.1 | 100/100 | 99.7/99.8 | 99.9/100 | 99.9/100 | 86.0/90.1 | 93.3/94.6 | 84.1/88.1 |
| ANGLE-I | RAISE* | 62.4/73.8 | 78.6/85.2 | 68.4/84.0 | 69.5/85.5 | 45.8/65.3 | 44.3/57.7 | 57.5/69.5 | 70.2/69.6 |
| | GENVP | 90.2/91.7 | 100/100 | 87.3/88.4 | 87.8/93.7 | 99.5/99.8 | 80.8/80.7 | 92.9/94.1 | 83.2/85.1 |
| COLOR-I | RAISE* | 68.0/71.8 | 99.7/99.9 | 97.7/98.7 | 98.4/99.3 | 17.1/28.0 | 7.6/17.4 | 74.1/80.6 | 81.4/78.6 |
| | GENVP | 69.2/74.5 | 94.9/98.2 | 93.5/97.6 | 93.8/98.6 | 18.0/33.9 | 10.8/17.3 | 91.6/91.4 | 82.0/84.6 |
| SIZE-I | RAISE* | 44.7/64.4 | 50.2/58.5 | 34.1/57.7 | 41.5/62.1 | 31.6/62.3 | 48.7/48.6 | 45.8/58.5 | 60.1/61.0 |
| | GENVP | 83.1/87.5 | 97.2/97.5 | 70.9/84.3 | 67.2/87.5 | 85.7/89.6 | 85.2/87.2 | 93.4/93.2 | 82.1/83.1 |
| SIZE-E | RAISE* | 28.5/43.6 | 39.1/52.5 | 24.5/45.4 | 26.5/47.7 | 19.5/35.7 | 17.2/31.6 | 33.1/47.7 | 39.4/44.7 |
| | GENVP | 45.0/65.5 | 72.3/83.3 | 48.1/62.2 | 45.0/66.7 | 35.8/53.9 | 35.7/51.0 | 78.4/74.2 | 68.3/67.1 |

**Baselines.** We compare our model puzzle-solving accuracy with the SOTA generative RPM approaches, including GCA (Pekar et al., 2020), ALANS (Zhang et al., 2022), PRAE (Zhang et al., 2021), and RAISE (Shi et al., 2024). We report their accuracy from Shi et al. (2024); for the RAISE model, we also reproduce their results for additional analysis and comparisons.

**Implementation Details.** We trained our model using the correct puzzle $\mathbf{X}$, the negative images $\mathbf{A}$, and the associated set of rules $\mathbf{R}$. We 'warm up' GenVP parameters by maximizing ELBO Eq. (6), and then continue their training combining ELBO and the contrastive terms in Eq. (11). In both cases, we used the AdamW algorithm (Loshchilov & Hutter, 2017) with a learning rate $10^{-4}$. Additional information for our experimental setup (i.e., hyperparameter values) can be found in Appendix B. Scalability and efficiency analysis of GenVP and baseline models in Appendix D.

## 4.1 RPM Solving

In this section, we evaluate the GenVP's RPM solving performance. Given a context matrix and the choice list (candidates), we ask the model to identify the best fit for the missing bottom-right images. We assess performance on both in-distribution and out-of-distribution examples.

### 4.1.1 In-Distribution Performance

Table 1 shows RPM solving accuracy for our GenVP and baseline models on RAVEN and I-RAVEN datasets. GenVP surpasses other methods across all layouts without needing specific image encoders or detailed knowledge of attribute relations, such as progression dynamics or arithmetic operations (e.g., addition, subtraction) as in PRAE and ALANs. GenVP's MoE module for rule prediction, its powerful contrastive scheme[1], and the modeling of the complete RPM reasoning process enables our approach to learn robust representations and reduce inference noise. This is in contrast to RAISE's structure, which is overly simplistic and sensitive to noise (i.e., error propagation due to faulty rule predictions). GenVP also significantly improves performance in complex grid configurations (O-IG, $2 \times 2$, $3 \times 3$), dealing with smaller objects and more intricate rules regarding number and position attributes. Finally, we compute the zero-shot performance of the RAVEN-trained models for the

---

[1]In Appendix C.5, we provide an ablation study comparing the training of GenVP with and without contrastive training, in order to understand the effect of our contrastive scheme.

RAVEN-FAIR dataset (see Appendix C.2). GenVP achieves a state-of-the-art 97.3% puzzle-solving accuracy.

### 4.1.2 OUT-OF-DISTRIBUTION GENERALIZATION

We evaluate models' OOD performance by creating new training sets using the following two procedures.

**Unseen Attribute Values.** We create new values for size, color and angle attributes via interpolation and extrapolation. The interpolation creates unseen intermediate states, and the extrapolation creates attribute states outside the original range. We provide the exact OOD values for each attribute in the Appendix C.1.

**Rule Exclusion.** We develop two OOD training sets following Shi et al. (2024), each omitting specific rules. The first, CS-HELD-OUT, excludes the *(Constant, Type)* rule from the C-S configuration. The second, O-IC-HELD-OUT, omits following rules from the O-IC configuration: *(Type In, Arithmetic)*, *(Size In, Arithmetic)*, *(Color In, Arithmetic)*, *(Type In, Distribute Three)*, *(Size In, Distribute Three)*, and *(Color In, Distribute Three)*. Models are trained on these sets and assessed using the original test set, which retained the excluded rules.

Table 2 shows the OOD performance on unseen attribute values for GenVP comparing with the SOTA RAISE model. GenVP outperforms RAISE in all setups, especially in OOD angle scenarios, where angle serves as a distracting attribute in RAVEN/I-RAVEN. By separating the latent space into relevant, $\mathbf{Z}_o$, and distracting, $\mathbf{Z}_{\bar{o}}$, components, GenVP minimizes the impact of such distractors. Conversely, RAISE struggles with puzzles that include unseen angles.

For the RPM-relevant attribute, such as color, OOD performance closely matches in-distribution across most configurations (C-S, L-R, U-D, $2\times2$, $3\times3$), demonstrating strong model generalization. However, in the O-IC, O-IG configurations, we observe a significant performance drop. This drop is due to poor color generalization in the outer objects of O-IC, O-IG layouts, which were consistently white during training but are replaced by gray tones in the OOD dataset. This deterministic training behavior restricts generalization in O-IC, O-IG, unlike in configurations like center-single, where object colors vary. However, our model still outperforms RAISE in O-IC and competes effectively in O-IG.

Our model generalizes well for interpolated values (SIZE-I) of the size attribute but shows poor generalization during size extrapolation (SIZE-E), as indicated by the worst performance in all OOD scenarios in Table 2. This is due to the inclusion of significantly smaller objects in size extrapolation compared to the in-distribution set, leading to reduced performance in O-IC, O-IG, and $3 \times 3$ layouts where GenVP encoders struggle to extract features from extremely small visual objects. However, compared to the RAISE model, GenVP still achieves superior performance across the board.

Table 3: OOD RAVEN/I-RAVEN puzzle solving accuracy (%) for unseen rules *RAISE\*: reproduced results*

| OOD | GENVP | RAISE* |
|---|---|---|
| CS-HELD-OUT | **100/100** | 97.0/98.8 |
| O-IC-HELD-OUT | **47.5/65.7** | 47.0/62.9 |

Table 3 presents results on OOD performance for unseen attribute rules for our model and the best-performing baseline model RAISE. Our model achieves a perfect generalization for the CS-HELD-OUT case, with the removal of the constant size rule from the training set. For the more challenging setup, O-IC-HELD-OUT, where a total of five rules are being removed, our GenVP still exceeds the SOTA performance.

### 4.1.3 SOLVING VISUAL PUZZLES WITH LARGE SPACES OF FEASIBLE SOLUTIONS

In Table 4, we present the puzzle-solving performance of generative approaches[2] for the challenging PGM and VAD having significantly larger solution spaces. We observe that RAISE is not able to adapt to the large solution space datasets PGM and VAD (especially for PGM) as it was discussed in Shi et al. (2024). In contrast, our approach, GenVP, demonstrates efficient generalization and

---

[2]PRAE and ALANS models depend on grid-structured RAVEN-based panels to crop individual objects. However, in PGM and VAD, there is no prior information on the location of each object. Consequently, PRAE and ALANS are not scalable for these datasets. GCA was evaluated in Pekar et al. (2020) on the PGM Neutral (N) and Interpolation (I) training sets, achieving a reported performance of 82.9% and 61.7%, respectively.

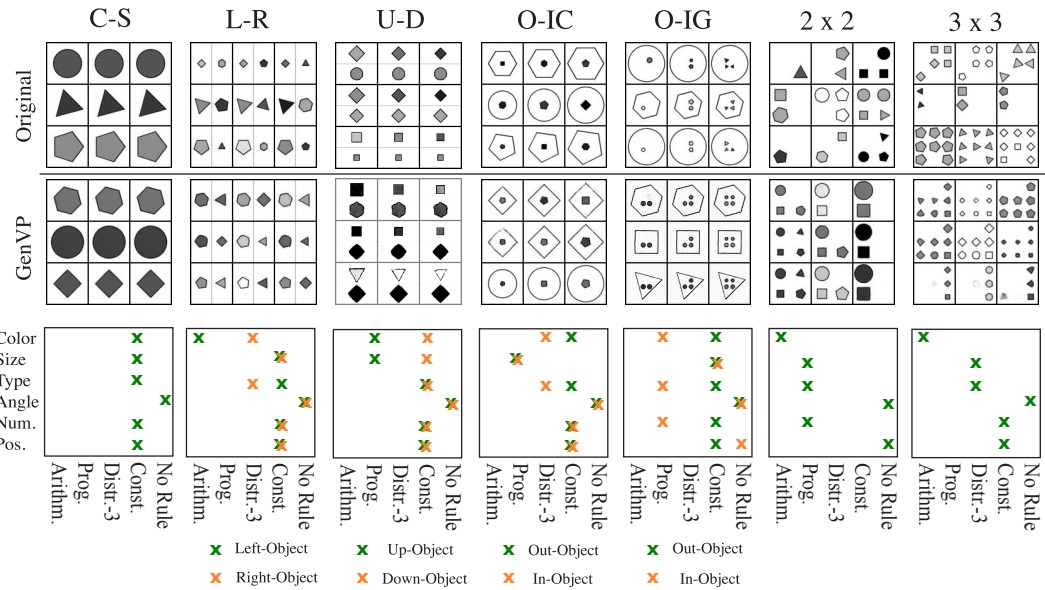

Figure 2: Generated RPM matrices by GenVP. Top: Original RPM matrices; Middle: GenVP generations; Bottom: Rules

achieves SOTA performance in almost all cases. This is a result of our robust mechanisms in both modeling and learning.

In particular, our MoE rule predictor extracts rule information from various puzzle views and multi-level features, encompassing low-level image features and high-level puzzle features, which increases the model robustness

Table 4: Puzzle-solving accuracy for the large solution space datasets PGM and VAD.

| DATASET | VAD | | | | | PGM | | | | | | | |
|---|---|---|---|---|---|---|---|---|---|---|---|---|---|
| CONFIG. | DT | I | E | TD-LT | TD-SC | N | I | E | LT | SC | AP | AR | ARP |
| RAISE | 75.7 | 87.6 | 60.7 | 63.8 | 60.0 | 22.4 | 15.3 | 13.1 | 15.2 | **18.1** | 19.0 | **18.3** | 19.4 |
| GENVP | **96.3** | **94.9** | **80.1** | **76.3** | **79.6** | **85.5** | **65.6** | **16.2** | **17.6** | 12.8 | **73.2** | 15.5 | **78.6** |

by reducing the prediction noise since we do not rely on a single predictor like prior work. Another crucial aspect of GenVP is its capability to filter out rule-irrelevant factors representations during the rule reasoning process, minimizing the impact of distractors and noise in the puzzle-solving process. In terms of learning robustness, cross-puzzle and cross-candidate contrastive learning encourages the model to learn potent features by comparing different valid or valid vs. invalid puzzles. This is critical for learning representations that can accommodate the big solution space present in PGM and VAD datasets. Refer to Appendix C.3 for more details.

## 4.2 RPM GENERATION

In this section, we evaluate the RPM generation capability of GenVP. We generate new RPM matrices $\hat{\mathbf{X}}$ using GenVP, given $\mathbf{R}$ as input, i.e., $\mathbf{R} \xrightarrow{\text{GenVP}} \hat{\mathbf{X}}$, and then perform a quantitative and qualitative analysis that examines the coherence and quality of the generation. Note that this generation process is different from the generation offered by RAISE, which focuses on the generation of the missing panels alone in the context of an incomplete RPM puzzle.

### 4.2.1 QUANTITATIVE ANALYSIS

**Experiment Setup.** We evaluate the coherence of GenVP generated RPM samples by assessing the accuracy of the predicted attribute rules $\hat{\mathbf{R}}$ compared to the input $\mathbf{R}$. We used the GenVP inference module to extract $\hat{\mathbf{R}}$, i.e., $\mathbf{R} \xrightarrow{\text{GenVP}} \hat{\mathbf{X}} \xrightarrow{\text{GenVP}^{-1}} \hat{\mathbf{R}}$ . We report the percentage of attributes consistent with the input rule across all generated RPMs for each layout configuration.

The GenVP inference model demonstrates a high level of accuracy in extracting attribute relations, as seen in Table 5, for the complete puzzles in the original test set (TEST ACC). This highlights

the model's ability to accurately deduce the rules of a valid puzzle, hence a reasonable choice for measuring coherence for the generated samples.

**Quantitative Results.** The quantitative results in Table 5 show that GenVP generates RPM samples with reasonable coherence, adhering to the rule matrix input. For simpler layouts such as C-S, L-R, U-D, and O-IC, GenVP produces highly coherent samples. Even with complex layouts involving more shapes, such as O-IG,

Table 5: GenVP inference model testing accuracy (row 1) and generated RPM coherence (row 2).

|  | AVG | C-S | L-R | U-D | O-IC | O-IG | 2 × 2 | 3 × 3 |
|---|---|---|---|---|---|---|---|---|
| TEST ACC | 99.8 | 99.5 | 99.8 | 99.9 | 99.9 | 99.0 | 97.4 | 97.0 |
| GEN. COHERENCE | 78.0 | 83.4 | 76.1 | 80.1 | 82.6 | 72.5 | 75.9 | 75.4 |

$2 \times 2$, and $3 \times 3$, it still scores relatively high. Overall, GenVP effectively understands and applies rules to generate coherent RPM instances. In Appendix C.4 we include additional quantitative analysis, including the performance for each specific rules with format (attribute, rule).

### 4.2.2 QUALITATIVE ANALYSIS

Fig. 2 presents examples created by GenVP, accompanied by the original RPM matrices for each layout. GenVP successfully generates novel RPM matrices that are distinct from the original ones while largely adhering to the same set of rules. Our model excels at forming shapes of various sizes, types, and colors, particularly in simpler layouts like C-S. Additionally, GenVP is capable of generating a diverse number of shape entities, as seen in the $3 \times 3$ configuration. Despite occasional challenges in generating high-quality outputs for smaller shapes, GenVP exhibits exceptional comprehension of rules, as evidenced by its ability to generalize them to novel RPM matrices. In Fig. 3, we investigate GenVP ability to generate multiple correct answers for a spe-

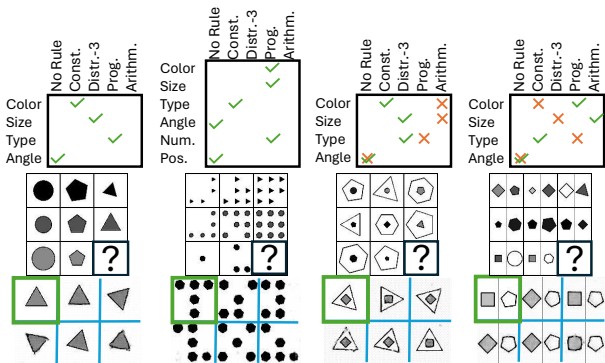

Figure 3: Multiple solutions for the bottom-right panel. Top: RPM rules, Middle: Context Matrix, Bottom: Solutions. Green-marked image is the dataset solution, followed by GenVP answers.

cific RPM panel, such as the bottom-right image. GenVP can generate various feasible answers by sampling different realizations of latent representations $\mathbf{Z}_o, \mathbf{Z}_{\bar{o}}$. In Fig. 3, we first present the solution provided from the dataset (image with green block), and seven GenVP generated answers for the bottom-right image. We include additional qualitative examples in Appendix H.

## 5 CONCLUSION

In this paper, we present GenVP, an advanced HVAE-based generative RPM solver. GenVP exceeds the SOTA benchmarks in terms of puzzle-solving accuracy in both the in-distribution and out-of-distribution scenarios within RAVEN-based datasets and efficiently generalizes to PGM and VAD characterized by enlarged feasible solution space. Furthermore, our model is also the first model to exhibit high-level rule-completion capability as demonstrated by the generation of novel RPM matrices. Despite surpassing the SOTA benchmarks, GenVP and similar RPM solvers face challenges with complex configurations like O-IG, $2 \times 2$, AND $3 \times 3$, as well as intricate arithmetic rules. Improving visual comprehension of numerical relationships is an ongoing challenge, and we reserved this for future work.

## 6 ACKNOWLEDGMENT

We thank the reviewers/AC for the constructive comments to improve the paper. HW is partially supported by Microsoft Research AI & Society Fellowship, NSF Grant IIS-2127918, NSF CAREER Award IIS-2340125, NIH Grant 1R01CA297832, and an Amazon Faculty Research Award.

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

## A GenVP Inference and Generative Model

**Inference Pipeline Variable Functional Form.** Starting from the puzzle-level representations $\mathbf{Z}, \mathbf{Z}_o, \mathbf{Z}_{\bar{o}}$ posteriors, we model them as conditionals Gaussian distributions:

$$q_\phi(\mathbf{Z}|\mathbf{X}) = \prod_{i\in[M]} \prod_{j\in[M]} \mathcal{N}\left(\mu_{\mathbf{z}_{ij},q}(\mathbf{x}_{ij};\phi), \sigma^2_{\mathbf{z}_{ij},q}(\mathbf{x}_{ij};\phi)I\right) \tag{12}$$

$$q_\phi(\mathbf{Z}_o|\mathbf{Z}) = \prod_{i\in[M]} \prod_{j\in[M]} \mathcal{N}\left(\mu_{\mathbf{Z}_o,q}(\mathbf{z}_{ij};\phi), \sigma^2_{\mathbf{Z}_o,q}(\mathbf{z}_{ij};\phi)I\right) \tag{13}$$

$$q_\phi(\mathbf{Z}_{\bar{o}}|\mathbf{Z}) = \prod_{i\in[M]} \prod_{j\in[M]} \mathcal{N}\left(\mu_{\mathbf{Z}_{\bar{o}},q}(\mathbf{z}_{ij};\phi), \sigma^2_{\mathbf{Z}_{\bar{o}},q}(\mathbf{z}_{ij};\phi)I\right). \tag{14}$$

In our implementation, similarly to the generation model, we reduce the model complexity by assigning the first $K_{Z_o}$ dimensions of $\mathbf{Z}$ to $\mathbf{Z}_o$ and the remaining ones to $\mathbf{Z}_{\bar{o}}$. Due to this modeling decision, we do not need to learn additional parameters for Eqs. (13) and (14).

**Generative Pipeline Variable Functional Form.**

$$p_{\boldsymbol{\theta}}(\mathbf{Z}_r|\mathbf{R}) = \mathcal{N}\left(\mu_{\mathbf{Z}_r}(\mathbf{R};\boldsymbol{\theta}), \sigma^2_{\mathbf{Z}_r}(\mathbf{R};\boldsymbol{\theta})\right) \tag{15}$$

$$p_{\boldsymbol{\theta}}(\mathbf{Z}_o|\mathbf{Z}_r) = \mathcal{N}\left(\mu_{\mathbf{Z}_o}(\mathbf{Z}_r;\boldsymbol{\theta}), \sigma^2_{\mathbf{Z}_o}(\mathbf{Z}_r;\boldsymbol{\theta})\right) \tag{16}$$

$$p_{\boldsymbol{\theta}}(\mathbf{Z}|\mathbf{Z}_o, \mathbf{Z}_{\bar{o}}) = \prod_{i\in[M],\, j\in[M]} \mathcal{N}\left([\mathbf{z}^T_{o,ij}, \mathbf{z}^T_{\bar{o},ij}]^T, \Sigma\left(\sigma^2_{\mathbf{Z}_o}, I\right)\right) \tag{17}$$

$$p_{\boldsymbol{\theta}}(\mathbf{X}|\mathbf{Z}) = \prod_{i\in[M]}\prod_{j\in[M]} \mathcal{N}\left(\mu_{\mathbf{x}_{ij}}(\mathbf{z}_{ij};\boldsymbol{\theta}), \sigma^2_{\mathbf{x}_{ij}}\right). \tag{18}$$

We simplify Eq. (17) by concatenating representations of active and inactive attributes for the mean vector and forming a block diagonal matrix $\Sigma(\cdot)$ for variance. For the image decoder in Eq. (18), we also simplify by setting the variance $\sigma^2_{\mathbf{x}_{ij}}$ to zero.

## B EXPERIMENTAL SETUP

**Data.** We assessed GenVP with the RAVEN-based datasets (RAVEN (Zhang et al., 2019a), I-RAVEN (Hu et al., 2021) and RAVEN-FAIR (Benny et al., 2021)), featuring seven layout types: C-S (centered single object), L-R (objects left and right), U-D (objects up and down), O-IC (nested single object), O-IG (nested $2 \times 2$ mesh), and $g \times g$ (objects in a $g \times g$ mesh for $g = 2, 3$). In valid puzzles, active object attributes include number, position, type, size, and color, with possible relations including constant, progression, arithmetic, and distribute-three; object orientation/angle and uniformity should be treated as distractors during testing. In the RAVEN-based datasets, the completed puzzles have dimensions $M = N = 3$, and the choice list has 8 images (one correct and seven incorrect). These datasets differ primarily in their choice list generation. The RAVEN dataset presents a more challenging choice list, with hard negative images differing from the target by only one attribute, yet a shortcut solution often exists through analysis of these images, hence the need for the I-RAVEN and RAVEN-FAIR to correct such a shortcut.

We conducted experiments using the challenging PGM and VAD datasets (Barrett et al., 2018; Hill et al., 2019). The PGM dataset includes seven different training and testing sets, with one for in-distribution and six for various out-of-distribution scenarios. The in-distribution set is labeled as Neutral (N), while the Interpolation (I) set uses out-of-distribution attribute values for testing by interpolating values of the training set regime. Similarly, the Extrapolation (E) set uses extrapolating attribute values for testing. Additionally, the datasets Attribute Shape Color (SC) and Line Type (LT) withhold the corresponding object attribute rules from the training dataset and test the out-of-distribution performance on puzzles with rules on shape color and line type.

The Held-Out Triples (Attr Rels (AR)) test involves holding out triples of (attribute, object, specific values) in the training set. The Held-Out Pairs of triples (Attr Rel Pairs (ARP)) and the Held-Out Attribute Pairs (Attr Pairs (AP)) are tests for compositional generalization. The ARP test set contains puzzles with the triples (attribute #1, object #1, value #1) and (attribute #2, object #2, value #2), whereas in training, there are no puzzles with this combination of triples. Similarly, the AP test set is designed to assess compositional generalization, where puzzles with the pairs (attribute #1, object #1) and (attribute #2, object #2) do not occur in the training set.

Each training set consists of 1.2 million puzzles, and each testing set consists of 200,000 puzzles. The complete puzzle dimensions are with $M = N = 3$ and the choice list has $A = 7$ incorrect images. Unlike RAVEN-based testing in PGM, all attributes can follow a rule or not. This means that an attribute for a puzzle can be a distractor, and for another, an attribute for which an active rule has to be inferred. This design choice results in a much larger puzzle solution space since the distractor attributes can have any value they want, and all images with different distractor values are possible solutions to the puzzle.

VAD puzzles focus on making analogies, which are accomplished by puzzles with two rows ($M = 2, N = 3$). The first row follows a rule of the type (rule_type #1, attribute #1, object #1, value #1), and the second row should also follow the same rule_type #1 but with different attributes or objects. Each puzzle has only one rule_type, which means that most of the object attributes are distractors. As a result, VAD has a much larger solution space than RAVEN-based datasets. However, it has a smaller choice list, with only four candidates (one correct and three incorrect ($A = 3$) generated by the learning by contrasting approach). VAD includes five different training and testing sets to

examine various types of OOD analogy-making. The training set consists of $600K$ examples, and the testing set consists of $100K$. Similar to PGM, it has Interpolation (I) and Extrapolation (E) sets for OOD attribute value generalization. It also includes a Domain Transfer (DT) set, which measures the model's ability to solve VAD puzzles with unfamiliar source domain (first puzzle row) to target domain (second puzzle row) transfers. Finally, similar to PGM, it evaluates generalization to unseen target domains for shape color (DT-SC) and line type (DT-LT).

**Model Architecture.** For the encoder from pixel space images to the latent dimension $\mathbf{z}_{ij}$ (EncoderZ) and the reverse decoder from $\mathbf{z}_{ij}$ to pixel space (DecoderX), we use the same architecture as those in RAISE. For the remaining encoders and decoders, we use fully connected neural networks. For our PGM and VAD models for the $\mathbf{Z}_o$ to $\mathbf{Z}_r$ mapping (encoder) we use instead of the sampled $\mathbf{Z}_o$ its attended representation using a transformer block with $MN$ tokens (for and PGM $M = N = 3$ and for VAD $M = 2, N = 3$), where each token is the $\mathbf{z}_{o,ij}$ latent representation for each image of the complete puzzle. In particular, for PGM, we have nine tokens, and in VAD, six. Our transformer has five heads and two layers.

Regarding the remaining dimensions of our variables, we set $H = W = 64$ for the image dimensions as in RAISE; for the intermediate latent variables, we have $K = 64, K_{Z_o} = 54, K_{Z_{\bar{o}}} = 10, K_{Z_r} = 192 = 3 \cdot 64, K_R = 12, N_R = 5$. For the $\beta$ hyperparameters we set them to $\beta_1 = 1, \beta_2 = 0, \beta_3 = 1, \beta_4 = 1, \beta_5 = 1, \beta_6 = 1, \beta_{R^*} = 250, \beta_G = 20, \beta_L = 20$. Notice that in our implementation, when rule annotations are available, we set $\beta_2 = 0$ to simplify our optimization objective since the reconstruction loss between rule annotations and predictions is enough to learn to extract the rules of a given complete puzzle. For PGM and VAD we use the following dimensions for the intermediate latent variables $K = 102, K_{Z_o} = 100, K_{Z_{\bar{o}}} = 2, K_R = 10, N_R = 6$. We also perform decoupling for the puzzle level latent variable $\mathbf{Z}_r$ (similarly to the image level decoupling), which has 300 rule relevant dimensions, 600 rule irrelevant for PGM, 200 rule relevant, and 400 rule irrelevant dimensions for VAD.

**Experimental Setting/Details.** We trained our models using the correct puzzle $\mathbf{X}$, the negative images set $\mathbf{A}$ available in the datasets ($A = 7$ for RAVEN-based and PGM and $A = 3$ for VAD), and the associated set of rules $\mathbf{R}$. We 'warm up' GenVP parameters by maximizing ELBO Eq. (6) and then continue their training combining ELBO and the contrastive terms in Eq. (11). We set the batch size to $B = \{$RAVEN-based: 100, PGM: 400, VAD: 400$\}$ RPM puzzles, which means that we use $B$ valid puzzles for ELBO and global contrasting and a batch size of $A = \{$RAVEN-based: 7, PGM: 7, VAD: 3$\}$ for the local contrasting loss. In both cases, we used the AdamW algorithm (Loshchilov & Hutter, 2017) with a learning rate $10^{-4}$.

**Experiments Compute Resources.** All the models are trained on a server with 24GB NVIDIA RTX A5000 GPUs, 512GM RAM, and Ubuntu 20.04. For the efficiency and scalability evaluations, we used a server with characteristics of 48GB NVIDIA RTX A6000 GPUs and Dual AMD EPYC 7352 @ 2.3GHz = 48 cores, 96 vCores CPU. GenVP is implemented with PyTorch.

## C EXPERIMENTS

### C.1 OUT-OF-DISTRIBUTION GENERALIZATION DETAILS FOR RAVEN-BASED DATASETS

**OOD attribute values.** Here, we include the details for reproducing the OOD attribute values in RAVEN and I-RAVEN datasets. The angle distractor initial values are $[-135, -90, -45, 0, 45, 90, 135, 180]$, for the OOD angle interpolating values we used the set $[-157, -112, -67, -22, 22, 67, 112, 157]$. For rule-relevant color attribute the in-distribution values are $[255, 224, 196, 168, 140, 112, 84, 56, 28, 0]$ and for our OOD analysis we used the interpolated values $[238, 210, 182, 154, 126, 98, 70, 42, 14]$. Finally, for the size attribute the in-distribution set is $[0.4, 0.5, 0.6, 0.7, 0.8, 0.9]$, our OOD values for interpolation analysis are $[0.45, 0.55, 0.65, 0.75, 0.85]$ and for extrapolation $[0.25, 0.35, 0.45, 0.55, 0.65, 0.75, 0.85, 0.95]$.

### C.2 RAVEN-FAIR DATASET ZERO-SHOT PERFORMANCE

In Table 6, we present the RAVEN-FAIR dataset (Benny et al., 2021) zero-shot performance using RAVEN dataset trained models. We notice that the proposed model, GenVP, achieves the best average puzzle-solving accuracy for the RAVEN-FAIR dataset. We could not train PRAE for the

Table 6: RAVEN-FAIR (Benny et al., 2021) zero-shot puzzle-solving accuracy using models trained on RAVEN dataset. *For PRAE, we do not report the* AVG *performance since we were not able to run the model for the* $3 \times 3$ *configuration because of PRAE's poor scalability, out-of-memory (OOM), even with batch size equal to one.*

| MODEL | AVG | CS | LR | UD | OIC | OIG | $2 \times 2$ | $3 \times 3$ |
|---|---|---|---|---|---|---|---|---|
| PRAE | - | **100.0** | **100.0** | 98.9 | 97.3 | 71.2 | 91.5 | OOM |
| RAISE | 95.6 | **100.0** | 99.9 | 99.8 | 99.2 | 94.5 | 86.0 | **90.0** |
| GENVP (OURS) | **97.3** | **100.0** | **100.0** | **100.0** | **100.0** | **95.7** | **96.1** | 89.2 |

$3 \times 3$ configuration because of its scalability problems (the model could not fit in our server even with a batch size equal to one).

## C.3 PUZZLE-SOLVING PERFORMANCE FOR DATASETS WITH LARGE SOLUTION SPACE

In (Shi et al., 2024), the authors discuss the challenges of generative approaches in abstract visual reasoning, particularly in puzzles with large solution spaces. These datasets differ from RAVEN-based datasets in the following ways: PGM and VAD have more complex panel layouts consisting of both geometric shapes and lines. Furthermore, the configuration layout is not predefined (i.e., a single object per image or four objects in a $2 \times 2$ grid), and objects can overlap. However, the most critical difference between these two types of datasets is the number of rules per puzzle. In RAVEN-based datasets, attributes like color, shape, and scale will always follow one of the active rules, so we know beforehand that these attributes follow a rule, and the quest is to find what is the type of the rule. In PGM and VAD datasets, few attributes follow a rule (usually one or two), so the model should also understand which attributes follow a rule and which are distractors. After that, it should reason about the rule type. Note that, the distractor attributes can have random values (i.e., if the shape is a distractor, it does not matter if the solution to the puzzle is a triangle, rectangle, pentagon, etc.), which drastically increases the solution space of an RPM puzzle.

In Table 4, we observe that RAISE is not able to adapt to the large solution space datasets PGM and VAD (especially for PGM) as it was discussed in (Shi et al., 2024). In contrast, our approach, GenVP, demonstrates efficient generalization to the challenging RPM and VAD datasets. This is a result of our robust mechanisms in both modeling and learning.

The MoE rule predictors extract rule information from various puzzle views and multi-level features, encompassing low-level image features and high-level puzzle features which increase the model robustness by reducing the prediction noise since we do not rely on a single predictor. Another crucial aspect of GenVP is its capability to filter out rule-irrelevant factors in the image-level ($Z$) and puzzle-level ($Z_r$) representations during the rule reasoning process, minimizing the impact of distractors and noise in the puzzle-solving process.

In terms of learning robustness, cross-puzzle and cross-candidate contrastive learning encourages the model to learn potent features by comparing different valid or invalid puzzles. This is critical for learning representations that can accommodate the big solution space present in PGM and VAD datasets.

Table 7: GenVP inference model Rule Estimation Testing Accuracy (row 1), Generated RPM coherence (row 2), and Multiple Solutions Generation coherence (row 3) in percentage (%) for RAVEN dataset.

| | AVG | C-S | L-R | U-D | O-IC | O-IG | $2 \times 2$ | $3 \times 3$ |
|---|---|---|---|---|---|---|---|---|
| TEST ACC | 99.8 | 99.5 | 99.8 | 99.9 | 99.9 | 99.0 | 97.4 | 97.0 |
| RPM GEN. COHERENCE | 78.0 | 83.4 | 76.1 | 80.1 | 82.6 | 72.5 | 75.9 | 75.4 |
| MULT. SOLS GEN. COHERENCE | 93.0 | 94.3 | 97.5 | 97.4 | 95.7 | 87.5 | 92.9 | 85.4 |

## C.4 RPM GENERATION PERFORMANCE

In this section, we provide additional quantitative results regarding the RPM generation performance of our model. In Table 7, we present the accuracy of rule prediction on the testing set for the following scenarios: a) ground truth complete puzzles (where we add the correct image in the bottom-right

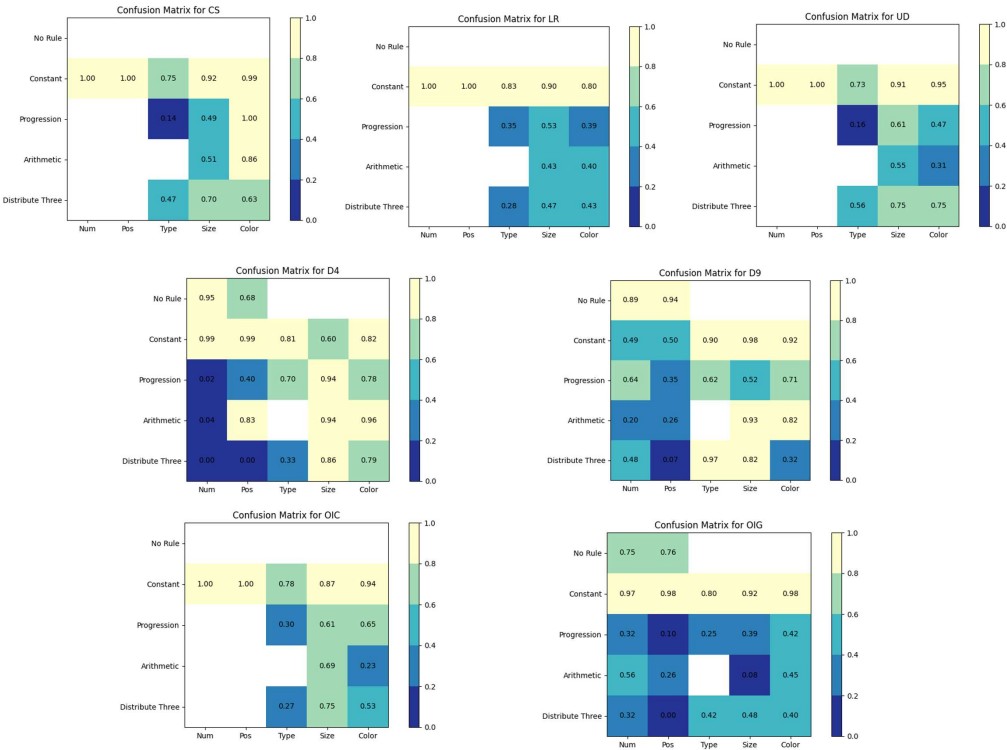

Figure 4: Rule Prediction performance for different (rule, attribute) pairs for RPM generated puzzles by GenVP trained on RAVEN-based datasets. The RPM puzzles using the GenVP generative graphical model (from rules to complete RPM puzzles).

position) in row 1 (labeled as TEST ACC); b) the RPMs generated from our model's generative graphical model (from rule matrix to RPM) in row 2 (labeled as RPM GEN. COHERENCE).

We notice that GenVP has good rule prediction for the testing set with ground truth complete puzzles, demonstrating that it correctly identifies the rules of a given complete RPM. Further, our generative model is also able to efficiently generate novel answers for all images in the RPM. Finally, for the challenging task of generating complete puzzles from an abstract rule matrix GenVP demonstrates effective generation capabilities by achieving an overall 78% rule accuracy prediction.

To gain a better understanding of the types of rule-attribute abstract relations that the model efficiently represents in the raw-pixel space, we analyze the overall rule prediction accuracy for different (attribute, rule) pairs. In Fig. 4, we can observe the performance analysis for generating RPMs from an abstract rule matrix for each layout and each rule relations. We find that our model faces the greatest challenge when understanding RPM puzzles with progression, arithmetic, and distribute-3 rules for number and position in grid configurations ($2 \times 2$ (D4) and $3 \times 3$ (D9)). However, for the $3 \times 3$ configuration, the performance of the number attribute column improves compared to the $2 \times 2$ configuration, which is expected due to the increased number of objects, leading to a wider variety in rule instantiations. This helps the model better understand the concept rules (progression, arithmetic, and distribute-3) related to number.

## C.5 TRAINING WITH ELBO AND CONTRASTIVE TERMS

In Tables 8 and 9, we present the result of GenVP when trained only with ELBO Eq. (6) and when trained with ELBO and the contrasting terms Eq. (11). The performance of GenVP improves dras-

Table 8: RPM solving accuracy (%) for RAVEN dataset. Training GenVP without contrasting terms (only ELBO maximization) (row 1) and with contrasting training (row 2).

| MODEL | AVG | C-S | L-R | U-D | O-IC | O-IG | 2×2 | 3×3 |
|---|---|---|---|---|---|---|---|---|
| GENVP (NO CONTRAST) | 50.5 | 54.5 | 48.7 | 48.9 | 49.3 | 46.0 | 59.0 | 46.9 |
| GENVP | **94.7** | **100** | **99.7** | **99.9** | **99.9** | **86.0** | **93.3** | **84.1** |

Table 9: RPM solving accuracy (%) for VAD and PGM datasets. Training GenVP without contrasting terms (only ELBO maximization) (row 1) and with contrasting training (row 2).

| MODEL | VAD | | | | | PGM | | | | | | | |
|---|---|---|---|---|---|---|---|---|---|---|---|---|---|
| | DT | I | E | TD-LT | TD-SC | N | I | E | LT | SC | AP | AR | ARP |
| GENVP (NO CONTRAST) | 67.0 | 68.0 | 55.4 | 54.7 | 58.5 | 47.4 | 30.3 | 14.3 | **17.8** | **12.7** | 27.4 | 16.1 | 26.6 |
| GENVP | **95.8** | **94.4** | **81.4** | **74.5** | **78.7** | **79.5** | **58.9** | **15.6** | 17.6 | **12.7** | **65.2** | **17.0** | 72.2 |

tically when we introduce contrastive learning. This is expected since ELBO only maximizes the likelihood of full puzzles, which are completed with the correct answer from the choice list; it is the Eq. (11) which introduces the negative candidates and helps GenVP distinguish the solution from hard negative choice list images.

Table 10: RAVEN-FAIR (Benny et al., 2021) puzzle-solving accuracy for different MoE estimators using GenVP individual rule predictions.

| MIXTURE TYPE | CS | LR | UD | OIC | OIG | 2 × 2 | 3 × 3 |
|---|---|---|---|---|---|---|---|
| WEIGHTED AVG | **100** | **100** | **100** | **100** | **95.7** | **96.1** | 89.2 |
| AVG | **100** | **100** | **100** | **100** | 95.2 | 95.1 | 88.7 |
| ARGMAX AVG | 99.9 | 97.4 | 97.5 | 99.4 | 85.2 | 75.8 | 68.8 |
| PROD | 97.5 | 96.2 | 97.1 | 99.1 | 91.7 | 81.2 | 83.3 |
| ARGMAX PROD | 97.6 | 93.4 | 91.5 | 95.4 | 84.9 | 68.2 | 68.3 |
| NN | **100** | 99.7 | 99.9 | 99.9 | 94.4 | 96.0 | **90.2** |

## C.6 ABLATION STUDY FOR DIFFERENT TYPES OF MIXTURE OF EXPERTS FOR RULE PREDICTION

When solving an RPM puzzle, we aim to exploit signals from all learned rule predictions (Eq. (1), Eq. (2), Eq. (3), Eq. (4)) to improve GenVPs' robustness. In our experiments, we analyzed the performance of the following mixture functions:

1. Non-Parametric:
    (a) WEIGHTED AVG: Our best-performing non-parametric function is the weighted average estimator, benefiting from the noise reduction properties of averaging (Kraftmakher, 2006):

$$\mathbf{p}_{\text{MoE},q} = \frac{1}{4}\left( p_{\mathbf{R}_1,q}(\mathbf{Z}_o) + \frac{1}{MN}\sum_{k\in[M],l\in[N]} p_{\mathbf{R}_2,q}\left(\mathbf{Z}_o \setminus \mathbf{z}_{kl,o}\right) + \frac{1}{M}\sum_{k\in[M]} p_{\mathbf{R}_3,q}\left(\mathbf{Z}_o \setminus \mathbf{z}_{k:,o}\right) + p_{\mathbf{R}_3,q}(\mathbf{Z}_r) \right). \tag{19}$$

    (b) AVG: All terms contribute the same:

$$\mathbf{p}_{\text{MoE},q} = \frac{1}{N_{\text{MoE}}}\left( p_{\mathbf{R}_1,q}(\mathbf{Z}_o) + \sum_{k,l} p_{\mathbf{R}_2,q}(\mathbf{Z}_o\setminus\mathbf{z}_{kl,o}) + \sum_k p_{\mathbf{R}_3,q}\left(\mathbf{Z}_o\setminus\mathbf{z}_{k:,o}\right) + p_{\mathbf{R}_3,q}(\mathbf{Z}_r) \right) \tag{20}$$

    where $N_{\text{MoE}}$ is equal to the total number of individual predictors.
    (c) ARGMAX AVG: First, we calculate the simple average of the individual rule predictors, as described in Eq. (20). Next, we replace each row of the return matrix with the one-hot encoding of the column index that has the maximum value.

Table 11: PGM (Barrett et al., 2018) and VAD (Hill et al., 2019) puzzle-solving accuracy for parametric (NN) and non-parametric (WEIGHTED AVG) MoE estimators using GenVP individual rule predictions.

| MIXTURE TYPE | VAD | | | | | PGM | | | | | | | |
|---|---|---|---|---|---|---|---|---|---|---|---|---|---|
| | DT | I | E | TD-LT | TD-SC | N | I | E | LT | SC | AP | AR | ARP |
| WEIGHTED AVG | 95.8 | 94.4 | **81.4** | 74.5 | 78.7 | 79.5 | 58.9 | 15.6 | **17.6** | 12.7 | 65.2 | **17.0** | 72.2 |
| NN | **96.3** | **94.9** | 80.1 | **76.3** | **79.6** | **85.5** | **65.6** | **16.2** | 17.6 | **12.8** | **73.2** | 15.5 | **78.7** |

    (d) PROD: The final rule matrix is the product of all rule predictors.

    (e) ARGMAX PROD: We begin by calculating the PROD mixture. Next, we replace each row of the return matrix with the one-hot encoding of the column index that has the maximum value.

2. Parametric (neural network approximators (NN)): We use a neural network to determine the best data-driven mixture of rule predictions, maximizing the puzzle-solving accuracy. The NN takes as input all rule predictions from the pretrained and frozen GenVP, which are arranged as channels of a CNN layer. Our objective is to use the rule predictions from the completed puzzles (using the choice list) to find the solution. We use standard cross-entropy loss to train NN.

Table 10 presents the puzzle-solving accuracy of RAVEN-FAIR using the aforementioned mixture functions. In almost all cases, WEIGHTED AVG achieves the best performance, followed by AVG and SC. In Table 11, we compute the puzzle-solving performance using the best non-parametric estimator (WEIGHTED AVG) and the parametric NN.

# D MODEL EFFICIENCY AND SCALABILITY EVALUATION

In this section, we measured generative models' efficiency and scalability abilities for abstract visual reasoning. For a fair comparison, we evaluate all models on the same server for the same batch size (set to 100). The server characteristics are 48GB NVIDIA RTX A6000 GPUs and Dual AMD EPYC 7352 @ 2.3GHz = 48 cores, 96 vCores CPU.

Table 12: Time and Memory requirements of GenVP and the baseline models RAISE (Shi et al., 2024), PRAE (Zhang et al., 2019b), and GCA (Pekar et al., 2020). All models were evaluated on the same server for a batch size of 100, except PRAE $3 \times 3$, which could not fit even with batch size one on our server (marked as out-of-memory (OOM)). Time per iteration complexity is computed across an average of 100 training iterations. The models that have configuration-independent architecture (therefore, architecture does not change for different configurations) are marked as N/A in the row CONFIG.

| MODEL
TRAINING PHASE
(FOR GENVP) | GENVP | | RAISE | PRAE | | | | | | | GCA |
|---|---|---|---|---|---|---|---|---|---|---|---|
| | ELBO | ELBO+C | | CS | LR | UD | OIC | OIG | $2 \times 2$ | $3 \times 3$ | |
| CONFIG | N/A | N/A | N/A | CS | LR | UD | OIC | OIG | $2 \times 2$ | $3 \times 3$ | N/A |
| # OF PARAMETERS | 8,879,877 | | 9,346,703 | | | | 245,555 | | | | 23,619,591 |
| BATCH SIZE | 100 | 100 | 100 | 100 | 100 | 100 | 100 | 100 | 100 | 1 | 100 |
| GPU MEM. (GB) | 2.8 | 5.17 | 4.2 | 1.12 | 1.76 | 1.76 | 1.76 | 3.64 | 3.03 | OOM | 15.3 |
| TIME/ITER. (MSEC) | 208.2 | 240.4 | 210.6 | 68.1 | 113.2 | 114.9 | 128.2 | 189.6 | 131.6 | - | 1082.7 |

## D.1 EFFICIENCY EVALUATION OF DIFFERENT ABSTRACT VISUAL REASONING MODELS

To evaluate the model's efficiency, we measure the average duration of a training iteration (time/iter. in msec). We compute the average of 100 training steps. We also record the GPU memory (in GB) requirements and the required number of parameters of each model. In Table 12, we present the efficiency analysis results. We note that all experiments were performed on a single GPU using the RAVEN training data. We notice that our approach, GenVP, compared to

- GCA: GenVP requires $\sim 1/3$ of the model parameters, $\sim 1/3$ of GPU memory, and $\sim 1/5$ of the iteration duration.
- PRAE: Although PRAE has significantly fewer model parameters than other generative models, due to the strong inductive biases used in its reasoning (PRAE uses knowledge of the rule formulas rather than learning them from the data), we observed that the model struggles to scale in complex scenarios. In particular, for the RAVEN configuration $3 \times 3$, PRAE could not even fit in our server with a batch size of one.
- RAISE: GenVP has $\sim 500,000$ *fewer* parameters from RAISE. The two models have similar GPU memory and average time per iteration requirements. Specifically, during the ELBO pretraining phase, GenVP occupies slightly less GPU and runs slightly faster. During our joint ELBO and contrasting training (ELBO+C), we observe the opposite: RAISE is slightly lighter and faster.

Table 13: Scalability evaluation for different GenVP neural network architectures and puzzle image sizes. All models were evaluated on the same server for a batch size of 100. Time per iteration complexity is computed across an average of 100 training iterations. We also report each model's accuracy on the RAVEN dataset (we train all models using all RAVEN configurations).

| IMAGE SIZE | $64 \times 64$ | $64 \times 64$ | $64 \times 64$ | $64 \times 64$ | $128 \times 128$ | $224 \times 224$ | $224 \times 224$ | $224 \times 224$ |
|---|---|---|---|---|---|---|---|---|
| ARCHITECTURE | CNN-S | CNN-L | RESNET | VIT | RESNET | RESNET | RESNET | RESNET |
| # OF PARAMETERS | 2,125,253 | 6,323,973 | 8,879,877 | 6,070,469 | 9,945,028 | 25,918,758 | 25,918,758 | 25,918,758 |
| BATCH SIZE | 100 | 100 | 100 | 100 | 100 | 100 | 100 | 100 |
| # OF GPUS | 1 | 1 | 1 | 1 | 1 | 1 | 2 | 4 |
| GPI MEM. (GB) | 3.49 | 7.02 | 5.17 | | 20.6 | 44.23 | 28.7/GPU | 14.9/GPU |
| TIME/ITER.(MSEC) | 209.6 | 268.1 | 240.4 | | 505.1 | 1263.9 | 729.0 | 390.3 |
| ACCURACY | 94.9 | 89.5 | 94.2 | 94.4 | 94.8 | 90.6 | | |

## D.2 GENVP SCALABILITY EVALUATION

In this section, we investigate the scalability properties of our model using different neural network architectures and sizes of the puzzle images. In particular, we perform experiments by training a single model for all RAVEN configurations for the following different architectures and puzzle image input sizes:

- For $64 \times 64$ image size
  - CNN-S: we use a much smaller CNN-based (LeCun et al., 1995) image encoder.
  - CNN-L: the architecture we used for training our RAVEN/I-RAVEN models in the main paper.
  - ResNet: we use a ResNet-based (He et al., 2016) image encoder.
  - ViT: we use a ViT-based (Dosovitskiy, 2020) image encoder.
- For $128 \times 128$ image size, we use a ResNet-based encoder architecture.
- For $224 \times 224$ image size, we use a ResNet-based encoder architecture.

Naturally, the time and memory complexity increase as we increase the model size and image resolution as shown in Table 13. Despite this, we found that even our heaviest version (with a $224 \times 224$ image) can still run on a single GPU on our server. Additionally, in order to reduce time complexity, we parallelized GenVP to run on multiple GPUs. This allowed our model to train faster on our server with smaller memory requirements per GPU at a time per iteration, which is comparable to our lowest resolution ($64 \times 64$) models. Regarding puzzle-solving performance, we noticed that our approach can be generalized to different architectures and achieve state-of-the-art performance with almost all setups having above 90% accuracy.

## E PERFORMANCE OF INDIVIDUAL RULE PREDICTORS VS. MIXTURE OF EXPERTS (MOE)

In this section, we compare GenVP's performance using individual rule predictors against its performance using the MoE (all predictors) for the RAVEN dataset. The results are summarized in Table 14. We observe the following:

- **State-of-the-Art Performance with MoE**: GenVP achieves the best performance (95.0% accuracy) when combining rule predictors through the MoE mechanism, demonstrating its effectiveness.

Table 14: In the MoE column, we present the puzzle-solving accuracy using all rule predictors. In the remaining columns, we show the accuracy when using individual rule predictors $R(\mathbf{Z})$, where $\mathbf{Z}$ denotes the input used for the individual rule prediction. For notation simplicity in the Table, for the rule predictions using only two rows, we use the notation $\mathbf{Z}_{ij}^{prow}$ meaning that we use the two rows $i$ and $j$, for $i, j = 1, 2, 3$. For the context-based rule predictions (one image is missing from the puzzle), we use the notation $\mathbf{Z}_{-i}^{ctx}$ meaning that the $i$-th image of the puzzle is missing for $i = 1, \ldots, 9$. The cases $\mathbf{Z}_{12}^{prow}$ and $\mathbf{Z}_{-9}^{ctx}$ are excluded since these instances do not contain the choice list/candidate images placed in the bottom-right position of the RPM matrix.

| MoE | $R(\mathbf{Z}_o)$ | $R(\mathbf{Z}_r)$ | $R(\mathbf{Z}_{13}^{prow})$ | $R(\mathbf{Z}_{23}^{prow})$ | $R(\mathbf{Z}_{-1}^{ctx})$ | $R(\mathbf{Z}_{-2}^{ctx})$ | $R(\mathbf{Z}_{-3}^{ctx})$ | $R(\mathbf{Z}_{-4}^{ctx})$ | $R(\mathbf{Z}_{-5}^{ctx})$ | $R(\mathbf{Z}_{-6}^{ctx})$ | $R(\mathbf{Z}_{-7}^{ctx})$ | $R(\mathbf{Z}_{-8}^{ctx})$ |
|---|---|---|---|---|---|---|---|---|---|---|---|---|
| 95.0 | 48.7 | 93.2 | 87.0 | 87.1 | 81.7 | 82.8 | 80.6 | 82.0 | 82.9 | 81.7 | 16.3 | 16.7 |

- **Performance of Individual Predictors**: Among individual predictors, the puzzle-level representation predictor $R(\mathbf{Z}_r)$ achieves the highest performance, followed by predictors using two puzzle rows ($R(\mathbf{Z}_{13}^{prow})$, $R(\mathbf{Z}_{23}^{prow})$) and context-based predictors ($R(\mathbf{Z}_{-1}^{ctx})$, $R(\mathbf{Z}_{-2}^{ctx})$, ..., $R(\mathbf{Z}_{-6}^{ctx})$). Predictors using image-level representations ($R(\mathbf{Z}_o)$) show comparatively lower performance.
  The high performance of $R(\mathbf{Z}_r)$ indicates that the puzzle-level latent variables exclude most of the information irrelevant to the rules (noise), making them robust for the puzzle-solving task.
  Additionally, we observe that the rule predictors focusing on only two rows outperform $R(\mathbf{Z}_o)$, which observes all three rows. By restricting these predictors to derive the rules from just two rows instead of three, the problem becomes more challenging, leading to greater robustness. This restriction prevents these predictors from exploiting shortcuts for rule prediction. For example, in the case of $R(\mathbf{Z}_o)$, the model might predict rules by focusing only on the first two rows of the puzzle, which would fail in scenarios where the third row contains an imposter image.
- **Challenging Scenarios**: The context predictors ($R(\mathbf{Z}_{-7}^{ctx})$, $R(\mathbf{Z}_{-8}^{ctx})$) perform poorly when the missing context matrix image lies in the same row as the negative candidate. This occurs because, in such cases, even though the last row contains an imposter image, the absence of the 7th or 8th image allows GenVP to imagine a plausible missing image that aligns with the puzzle rule.
  For example, consider a puzzle governed by an arithmetic rule based on the number of objects. The correct puzzle might have a number of objects (1,2,3; 2,2,4; 3,2,5), where the third image in each row is the sum of the first two. Now, suppose the last row contains an incorrect image, such as (1,2,3; 2,2,4; 3,2,4). This row violates the arithmetic rule, and GenVP should identify this as a puzzle with an incorrect candidate.
  However, if we consider the $R(\mathbf{Z}_{-7}^{ctx})$ predictor and observe a partially complete matrix, such as (1,2,3; 2,2,4; 3,-,4), this puzzle could still be valid under the arithmetic rule if the missing image were (1,2,3; 2,2,4; 3,1,4). Thus, when the 7th or 8th image is removed from a puzzle containing an imposter image in the 9th panel, GenVP can imagine a plausible image that corrects the rule violation, leading to a performance drop for these context predictors.

Table 15: GenVP Puzzle Solving Accuracy for the RAVEN dataset. We train four different versions, using different hyperparameter values $\beta_i, \beta_G, \beta_L, \beta_{R^*}$ for $i = 1, \ldots, 6, i \neq 2$.

| GENVP HYPERPARAMETERS | | | | | | | | ACCURACY |
|---|---|---|---|---|---|---|---|---|
| $\beta_1$ | $\beta_{R^*}$ | $\beta_3$ | $\beta_4$ | $\beta_5$ | $\beta_6$ | $\beta_G$ | $\beta_L$ | |
| 1 | 250 | 1 | 1 | 1 | 1 | 20 | 20 | 94.7 |
| 1 | 1 | 1 | 1 | 1 | 1 | 20 | 20 | 95.0 |
| 1 | 1 | 1 | 1 | 1 | 1 | 1 | 1 | 92.1 |
| 1 | 1 | 1 | 1 | 1 | 1 | 40 | 40 | 94.9 |

# F   GENVP PERFORMANCE VS. HYPERPARAMETERS VALUES

## F.1   ABLATION ON DIFFERENT OPTIMIZATION OBJECTIVE HYPERPARAMETERS

We conduct additional ablation studies to evaluate the influence of hyperparameters on GenVP's performance. Our results, in Table 15 show that GenVP performance remains robust across a range of configurations. In particular:

- **Rule Matrix Weight** ($\beta_{R*}$): In our primary experiments, we assigned a higher weight ($\beta_{R*} = 250$) to the rule matrix prediction term compared to other ELBO terms ($\beta_i = 1, i \neq 2$). To test sensitivity, we set all $\beta_i$ and $\beta_{R*}, i = 1, 2, \ldots, 6, i \neq 2$ to one, effectively removing the emphasis on the rule matrix. This resulted in an improvement in performance (94.7% to 95.0%) demonstrating that the model is robust even when the weighting is altered.
- **Contrastive Loss Weights** ($\beta_G, \beta_L$): In our main experiments, $\beta_G$ and $\beta_L$ were set to 20. In the additional experiments, we evaluated three configurations ($\beta_G, \beta_L = 1, 20, 40$). Results showed that very small weights ($\beta_G, \beta_L = 1$) caused a modest degradation (92.1% accuracy), but increasing the weights ($\beta_G, \beta_L = 20, 40$) stabilized performance at 95%. This suggests that GenVP benefits from reasonable tuning of contrastive terms but does not require fine-grained optimization.
- **Performance Consistency Across Hyperparameters**: Across all tested hyperparameter configurations, GenVP maintained strong puzzle-solving accuracy. The observed variations in performance were within an acceptable range, indicating that the model's core design, alongside its generative and contrastive training schemes is robust to hyperparameter changes.

In summary, while adjustment of certain hyperparameters, such as $\beta_{R*}$, $\beta_G$, and $\beta_L$, can optimize performance, GenVP does not rely heavily on precise tuning. Its stability across a wide range of settings underscores its adaptability and reliability in RPM puzzle solving tasks.

Table 16: GenVP puzzle solving accuracy for the RAVEN dataset. We trained five different versions of the model, using varying hyperparameter values $K$, $K_{Z_o}$, $K_{Z_{\bar{o}}}$, $K_{Z_r}$, $K_R$, and $N_R$, which control the size of GenVP latent variables.

| # PARAMETERS | $K$ | $K_{Z_o}$ | $K_{Z_{\bar{o}}}$ | $K_{Z_r}$ | $K_R$ | $N_R$ | ACCURACY |
|---|---|---|---|---|---|---|---|
| 8,250,313 | 32 | 20 | 12 | 60 | 12 | 5 | 91.6 |
| 8,912,077 | 64 | 54 | 10 | 162 | 22 | 5 | 95.3 |
| 8,957,157 | 64 | 54 | 10 | 162 | 12 | 15 | 92.4 |
| 9,758,185 | 128 | 100 | 28 | 300 | 12 | 5 | 94.8 |
| 10,208,245 | 128 | 126 | 2 | 378 | 12 | 5 | 94.9 |

## F.2   ABLATION ON VARYING LATENT VARIABLES DIMENSIONALITY

In this ablation study, we conducted additional experiments to explore the impact of latent variable size on GenVP's performance. As observed in Table 16, GenVP consistently achieves high accuracy across a range of hyperparameter configurations. Notably, if the latent variable dimensions and total number of model parameters are too restrictive (do not have enough capacity to capture the puzzle complexity), we expect GenVP's performance to be affected. However, although representing each pixel-level image of RAVEN with a latent vector $\mathbf{z}_{ij}, i, j \in 1, 2, 3$ of dimensionality $K = 32$ is restrictive (prior work, i.e. RAISE, uses 64-dimensional image encodings in the latent space), GenVP manages to achieve an accuracy of 91.6% (first row in Table 16). As the model total parameters and the capacity of the latent space vectors increase, GenVP's accuracy improves and stabilizes at $\sim 95\%$. This indicates that GenVP is robust to architectural hyperparameter choices, though models with large enough latent vectors generally yield better results.

Table 17: GenVP performance when trained without rule annotations for the RAVEN and I-RAVEN datasets.

| DATASET | RAVEN | I-RAVEN |
|---|---|---|
| RAISE | 54.5 | 67.7 |
| GENVP | 70.3 | 73.8 |

## G    GENVP WITHOUT RULE ANNOTATIONS

We investigated the performance of GenVP without any rule annotations, using only the puzzle images and their correct candidate label. The results in Table 17 highlight GenVP's ability to perform reasonably well without explicit rule annotations, achieving significantly higher accuracy than in prior work. This indicates that GenVP captures useful latent representations even without direct supervision of the rules.

## H    RPM GENERATION QUALITATIVE EXAMPLES

In Figs. 5 to 8 we include additional generated RPMs from GenVP. The generation process starts by selecting a desired set of rules and then performing ancestral sampling for the rule-invariant latent representation $\mathbf{Z}_{\bar{o}}$. In each case, we present three different generated RPM samples for a specific set of rules (located in the top part of the figures).

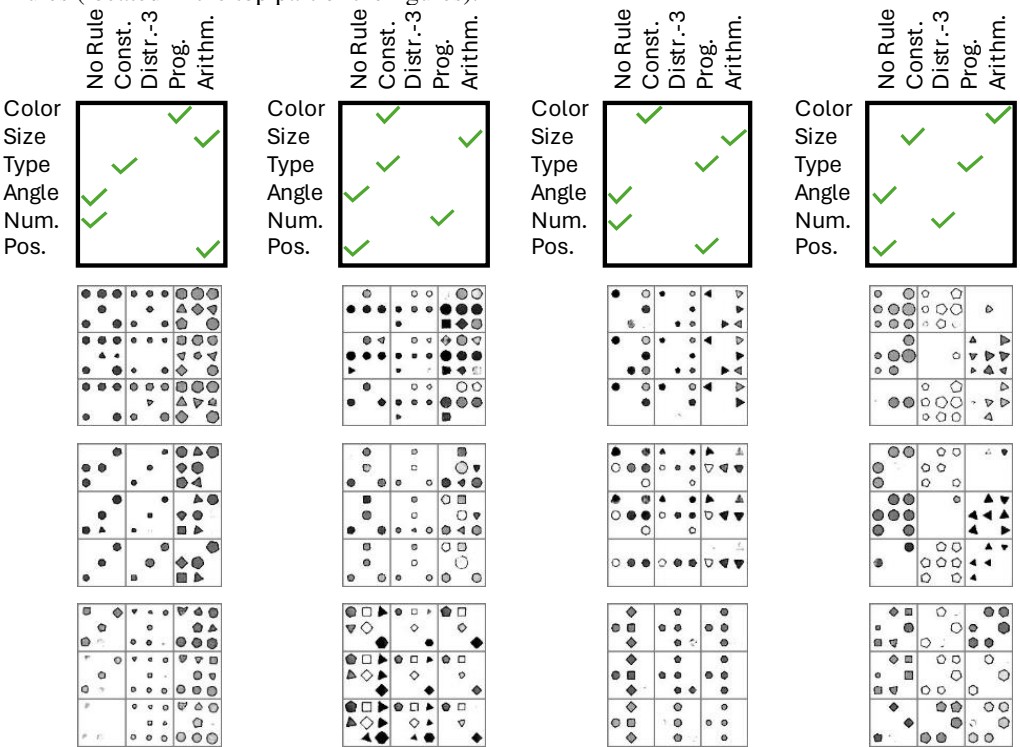

Figure 5: Generated RPM matrices by GenVP for distribute nine $(3 \times 3)$ configuration. Top: Rules; Bottom: Three different sampled generated puzzles.

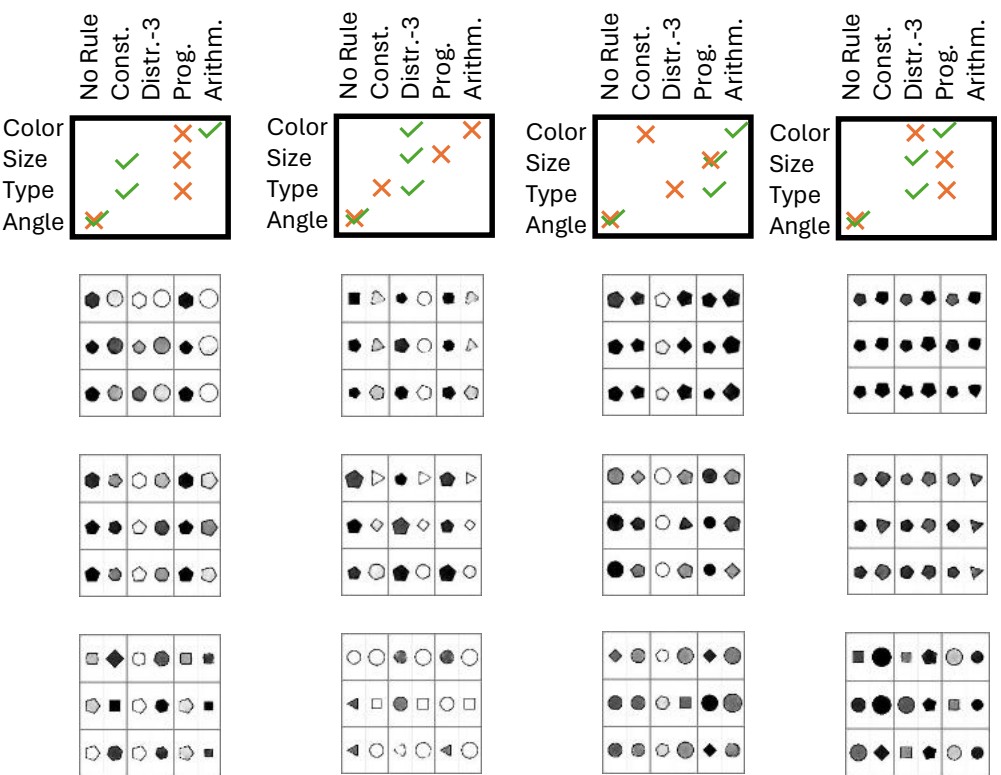

Figure 6: Generated RPM matrices by GenVP for L-R configuration. Top: Rules; Bottom: Three different sampled generated puzzles.

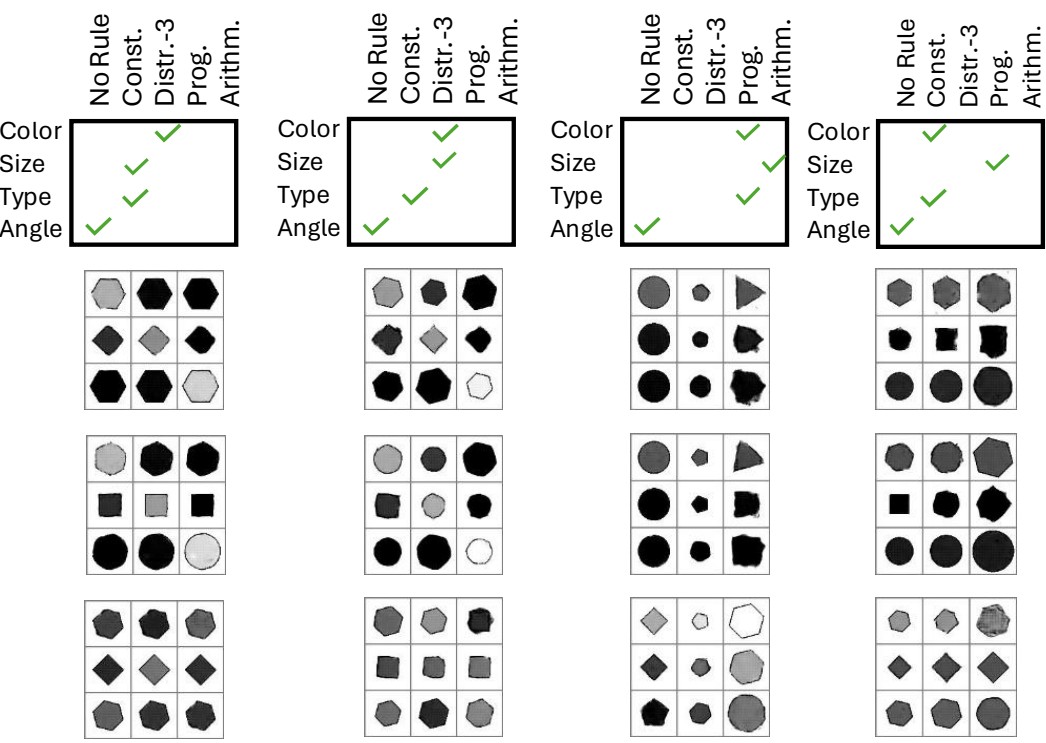

Figure 7: Generated RPM matrices by GenVP for center single (c-s) configuration. Top: Rules; Bottom: Three different sampled generated puzzles.

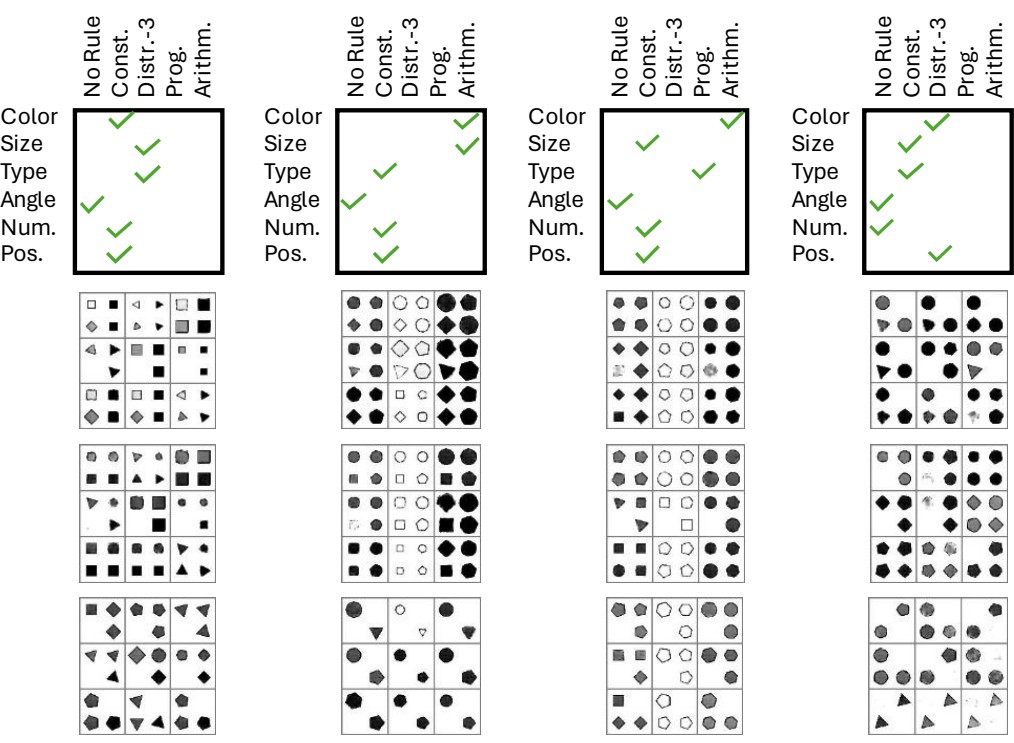

Figure 8: Generated RPM matrices by GenVP for distribute four ($2 \times 2$) configuration. Top: Rules; Bottom: Three different sampled generated puzzles.

