# OpenReview forum: "GenVP: Generating Visual Puzzles with Contrastive Hierarchical VAEs"
_ICLR.cc/2025/Conference — ICLR 2025 Poster_

### Official Review · Reviewer_npYP · 2024-11-01

**Soundness:** 3
**Presentation:** 3
**Contribution:** 3
**Rating:** 6
**Confidence:** 3

**Summary:**

This article introduces a new framework called Generative Visual Puzzles (GenVP), which aims to simulate the entire generation process of Raven's Progressive Matrices (RPMs). It not only performs excellently in solving existing puzzles but also demonstrates strong capabilities in creating new puzzles and generalizing to new, unseen puzzle scenarios.

**Strengths:**

1.The authors propose GenVP, a novel approach for solving and creating visual puzzles.
2.The authors design a new cross-puzzle and cross-candidate contrastive loss for AVR. The proposed GenVP is robust to noise.
3.The authors conducted extensive experiments to demonstrate the superior performance of the GenVP method.
4.GenVP can generate a large number of new puzzles beyond the original source dataset.

**Weaknesses:**

1.Although the author's experiments are very detailed, the rules of the puzzle generation task in this article are relatively simple, and the task's search space is relatively small.
2.According to Figure 2, it can be seen that the image quality of the puzzles generated by this method is still not good enough; the edges of the puzzle elements are relatively blurry, and there are artifacts.

**Questions:**

1.Although the author has done a lot of mathematical derivations, I am still somewhat confused about the process of generating puzzles. Could the author please describe in more detail how to generate puzzles from the rules?
2.It seems that the author did not compare the visualization of image generation with previous methods. Is it because previous methods couldn't generate such images?
I will consider to raise my score according to the rebuttal and discussion with  other reviewers

---

> ### Author Response · Authors · 2024-11-23
> **Official Author Response (Part A)**
>
> ### Response to Weakness 1
> > Although the author's experiments are very detailed, the rules of the puzzle generation task in this article are relatively simple, and the task's search space is relatively small.
>
> Thank you for mentioning this.
>
> **Existing Methods Struggle in These Tasks:** Actually, while the rules of RPM settings might appear simple to humans, it is far from straightforward for machines to process and reason about these rules effectively. Previous studies [2], [3] have shown that AI models often struggle to understand arithmetic or logical rules embedded within RPM tasks, highlighting the challenges posed by even seemingly simple rules.
>
> **Large Search Space:** Additionally, while the number of rules in datasets like RAVEN may seem small (e.g. four in RAVEN), their combination with other factors, such as multiple objects (up to nine in RAVEN), numerous attributes (more than six in RAVEN), and varied attribute values (ten different colors in RAVEN)—leads to huge puzzle spaces. **This leads to a total of 440,000 rule annotations** in RAVEN [1]. For PGM, this number is even larger; consider only that each training set (eight in total) alone contains over one million different puzzles.
>
> **Compounding Complexity:** It is also important to note that RPM problems involve a context matrix and a choice list. Together, these components interact with the rules, attributes, and values, making the problem space significantly larger and more intricate than it might initially seem. This complexity becomes even more apparent in datasets such as PGM and VAD, where only one attribute typically follows a rule, while others act as distractors. This distractor mechanism drastically increases the solution space and often proves detrimental to previous state-of-the-art models like RAISE, which struggle to handle such complexities effectively.
>
> Finally, while exceeding the state-of-the-art performance in in-distribution settings, we also demonstrated that our method surpasses the state-of-the-art RAISE model in out-of-distribution scenarios. By testing our method on large solution space datasets like PGM and VAD, we showed superior performance (Section 4.1.3). We hope this explanation clarifies the concern and highlights our contributions to advancing research on the AVR problem.
>
> [1] Chi Zhang, Feng Gao, Baoxiong Jia, Yixin Zhu, and Song-Chun Zhu. Raven: A dataset for relational and analogical visual reasoning. In Proceedings of the IEEE/CVF conference on computer vision and pattern recognition, pp. 5317–5327, 2019a.
>
> [2] Hersche, M., Camposampiero, G., Wattenhofer, R., Sebastian, A., & Rahimi, A. Towards Learning to Reason: Comparing LLMs with Neuro-Symbolic on Arithmetic Relations in Abstract Reasoning.
>
> [3] Małkiński, M., & Mańdziuk, J. (2023). A review of emerging research directions in abstract visual reasoning. Information Fusion, 91, 713-736.
>
> ### Response to Weakness 2
> > According to Figure 2, it can be seen that the image quality of the puzzles generated by this method is still not good enough; the edges of the puzzle elements are relatively blurry, and there are artifacts.
>
> Thank you for mentioning this.
>
> Firstly, we would like to emphasize that GenVP is the first practical generative model capable of creating complete RPM puzzles, a significant advancement over existing methods that lack such generative capabilities.
>
> Improving the image quality of generated puzzles is an important direction for enhancing GenVP's overall performance. One way to enhance the image quality is by using a more powerful architecture (e.g., increasing the number of layers or parameters), utilizing higher-resolution images, or continuing training to further reduce the image reconstruction loss. In our experiments, we selected the checkpoint with the first best puzzle-solving accuracy for evaluation. However, GenVP rapidly achieves state-of-the-art accuracy, which remains consistent in subsequent training iterations.
>
> To address the reviewer's concern, we repeated our experiments and retained the last checkpoint that achieved the best puzzle-solving accuracy while also minimizing the image reconstruction loss. This checkpoint maintains state-of-the-art performance in puzzle-solving accuracy while producing sharper images with fewer artifacts.
>
> We will update the figures in the main paper with these improved results to better reflect the enhanced image quality.

---

> > ### Author Response · Authors · 2024-11-23
> > **Official Author Response (Part B)**
> >
> > ### Response to Question 1
> > > Although the author has done a lot of mathematical derivations, I am still somewhat confused about the process of generating puzzles. Could the author please describe in more detail how to generate puzzles from the rules?
> >
> > We are sorry for the confusion and are glad to explain the puzzle generation process in more detail. GenVP generates puzzles step-by-step from a given set of rules as follows:
> >
> > 1. **Select the Desired Rules**: First, we create a rule matrix, $\mathbf{R}$, that encodes the specific rules we want the final puzzle to follow.
> >
> > 2. **Generate Puzzle-Level Representation ($\mathbf{Z}_r$)**: We use the learned decoder to map the rule matrix, $\mathbf{R}$ (the decoder input), to the mean and variance of a Gaussian latent representation, $\mathbf{Z}_r$, at the puzzle level. From this Gaussian distribution, we then sample a specific $\mathbf{Z}_r$ value.
> >
> > 3. **Generate Image-Level (Rule-Relevant) Representation ($\mathbf{Z}_o$)**: Using the sampled $\mathbf{Z}_r$ from Step 2, we apply the learned decoder to produce the mean and variance of the image-level Gaussian latent variables, $\mathbf{Z}_o$. We then sample a value of $\mathbf{Z}_o$ from this distribution, which provides a rule-relevant representation for each puzzle image.
> >
> > 4. **Sample Rule-Irrelevant Factors ($\mathbf{Z}_{\bar{o}}$)**: For any factors that are not directly related to the rule matrix, we sample values directly from a standard normal distribution to capture puzzle variability unrelated to the specific rules.
> >
> > 5. **Generate Pixel-Level Puzzle Images**: Finally, we combine/concatenate the samples $\mathbf{Z}\_o$ and $\mathbf{Z}\_{\bar{o}}$ into a single representation, $\mathbf{Z}$, at the image level. Then $\mathbf{Z}$ will be the input to the corresponding learned decoder, which will generate the actual pixel-level images of the puzzle, denoted as $\mathbf{X}$.
> >
> > This step-by-step process enables GenVP to create puzzles that follow the chosen rules while incorporating some randomness in rule-irrelevant aspects, producing varied yet consistent puzzle images.
> >
> > _Note: Each step has a different decoder/neural network._
> >
> > ### Response to Question 2
> > > It seems that the author did not compare the visualization of image generation with previous methods. Is it because previous methods couldn't generate such images?
> >
> > Thank you for your question. Yes, GenVP is the first model to explore the generative task of creating complete puzzles. Previous methods were designed to solve existing puzzles rather than generate them, and they lack the necessary generative capabilities. As a result, there are no comparable visualizations from prior work for this task. This highlights the unique contribution of GenVP in advancing generative modeling for abstract reasoning tasks.

---

> > > ### Comment · Reviewer_npYP · 2024-11-25
> > >
> > > I appreciate all the authors for their detailed response. Their feedback has addressed most of my concerns. As of now, I am leaning towards keeping my original score but will wait for additional reviewer discussion to update my score.

---

> ### Author Response · Authors · 2024-11-25
>
> We are delight to find our response address the reviewer's concern. However, we do wish to ask if there is further concerns and comments that prevent the reviewer from changing their score if they find the concerns addressed. Since there is still time left for the discussion period and the authors will have one more day (Nov. 27) to response after the open discussion phase end, we would really appreciate the reviewer to give us further feedback on any remaining questions that is preventing the reviewer from updating the score and gives us the chance to clarify them again for you. We thank the reviewer again for their work and hope to hear from you soon.

---

> > ### Author Response · Authors · 2024-12-01
> > **Kindly Reminder**
> >
> > Dear Reviewer npYP,
> >
> > Thank you for your review and engagement during the discussion period.
> >
> > We would like to inform you that we have updated Figure 2 in the main paper, incorporating examples generated using the latest and best GenVP checkpoints with enhanced image quality.
> >
> > As the ICLR Discussion Period is concluding soon (Dec. 2nd (AOE) for reviewers and Dec. 3rd (AOE) for authors), we kindly seek your feedback on whether our responses address your concerns or if there are any additional questions or suggestions you would like us to address.
> >
> > Thank you once again for your time!
> >
> > Best,
> >
> > The Authors

---

### Official Review · Reviewer_GBgR · 2024-11-02

**Soundness:** 3
**Presentation:** 3
**Contribution:** 2
**Rating:** 5
**Confidence:** 4

**Summary:**

This paper presents a latent variable model named GenVP for RPM-style visual puzzles. GenVP is trained from RPM images and the corresponding categorical rule matrices by inferring hierarchical Gaussian latent representations, which can be decoded back to images, through an ELBO loss and a global-local masked contrastive loss. GenVP learns a set of underlying rules controlling how an RPM is composed. The experiment results show that GenVP outperforms SOTA generative approaches in both out-of-distribution generalization and large solution space scenarios.

**Strengths:**

1.	This paper defines delicate generation and inference models for RPMs, and the details are easy to follow. The choice and hierarchy of latent variables are reasonable, e.g., decomposing relevant and irrelevant attributes from an image.
2.	Experiments are comprehensive, with a focus on OOD configurations and some challenging datasets. According to the testing and visualization results, the proposed GenVP outperforms SOTA generative approaches.

**Weaknesses:**

1.	Limitation of the application scenarios. The design of the generative model seems rather tailored to RPM problems. This focuses on a very specific problem hence its significance might be limited. It is unclear how to apply GenVP in other abstract visual reasoning tasks.
2.	Requirement of annotation of rules. Can a GenVP be trained without supervision of rules? For human beings, one does not require to learn rules through supervision, the rules can be discovered by the test subject.
3.	The interpretability of answer selection. GenVP infers MoE rule matrix predictions for candidates and chooses the one with the largest set of active rules as the final answer. Why not select answers by comparing the generated answers to candidates? Maybe it is a more human-like way of answer selection.

**Questions:**

1.	In the graphical model of GenVP in Figure 1, $R$ is a shaded circle which means observation; however, from the context of model description it seems that $R$ is inferred.
2.	Could the authors provide results on any other visual puzzles or abstract visual reasoning tasks? My main concern is whether GenVP can be applied to more realistic reasoning problems with a unspecific rule set.

---

> ### Author Response · Authors · 2024-11-23
> **Official Author Response (Part A)**
>
> ### Response to Weakness 1
> > Limitation of the application scenarios. The design of the generative model seems rather tailored to RPM problems. This focuses on a very specific problem hence its significance might be limited. It is unclear how to apply GenVP in other abstract visual reasoning tasks.
>
> This is a good question.
> We would like to reiterate that
> + The RPM problem is a solid benchmark that examines the machine's abstract visual reasoning capability.
> + We have tested our GenVP model across multiple datasets with different settings, therefore demonstrating its generality.
> + Our method has been shown to be superior in the in-domain experiments. Our generative approach also helps us to outperform state-of-the-art methods on the out-of-distribution settings (Table 2), e.g., compared to the RAISE model (ICLR 2024).
> + We have also provided experiments that showcase how our approach can scale up to puzzles with large solution spaces.
> + Adapting the AVR to direct application is indeed a valid and important research topic; yet there is still room to improve the generalizability of these AVR models within the RPM framework. Our research, therefore, leans toward the latter topic, and we have demonstrated through our results that we made a significant contribution with our proposed GenVP structure.
>
> ### Response to Weakness 2
> > Requirement of annotation of rules. Can a GenVP be trained without supervision of rules? For human beings, one does not require to learn rules through supervision, the rules can be discovered by the test subject.
>
> Thank you for this insightful question. While humans acquire the skills to discover rules without explicit supervision, learning-based generative models like GenVP face challenges in achieving similar levels of abstraction and generalization without guidance. Rule annotations serve as a crucial component for training AVR models like GenVP by providing the structure needed to effectively learn the generative process.
>
>
>
> - **Rule Annotations Enable Controlled Generation:** Rule annotations prompts GenVP to generate puzzles governed by specific rules, ensuring structured and interpretable puzzle generation. This aids in both the explainability of the model and its ability to reliably adhere to desired rules in generated puzzles.
>
> - **Consistency with Prior Methods:** As with other state-of-the-art models in this domain (e.g., GCA, ALANS, PRAE, RAISE), GenVP relies on rule annotations for effective training. This reliance aligns with current approaches that use supervision to achieve high performance.
>
> - **Challenges of Unsupervised Learning:** Training GenVP without rule annotations would require the model to infer rules solely from visual patterns, a highly challenging task for AI models. While humans excel at this due to innate reasoning abilities and prior knowledge, GenVP benefits from the explicit guidance provided by annotations.
>
> Inspired by your comments, we conducted an ablation study where we excluded rule annotations during GenVP training. **(Rebuttal) Table 2** below shows the results. While performance decreased, GenVP still outperforms the existing model RAISE in the rule annotation-free scenario.
>
> _**(Rebuttal) Table 2**: GenVP puzzle-solving accuracy when trained without rule annotations._
>
> | Dataset                     | RAVEN | I-RAVEN |
> |-----------------------------|-------|---------|
> | RAISE (no rule annotations) | 54.5  | 67.7    |
> | GenVP (no rule annotations) | 70.3  | 73.8    |
>
> These findings underscore *GenVP's potential to address unsupervised learning challenges* in visual reasoning tasks. Despite certain limitations observed on RPM datasets, the substantial improvement achieved by GenVP over previous approaches marks a significant step toward reducing dependence on rule annotations. This progress opens an exciting avenue for future research, focusing on developing models capable of human-like reasoning without requiring explicit supervision.

---

> > ### Author Response · Authors · 2024-11-23
> > **Official Author Response (Part B)**
> >
> > ### Response to Weakness 3
> > > The interpretability of answer selection. GenVP infers MoE rule matrix predictions for candidates and chooses the one with the largest set of active rules as the final answer. Why not select answers by comparing the generated answers to candidates? Maybe it is a more human-like way of answer selection.
> >
> > Thank you for raising this interesting point. We propose an inference algorithm based on puzzle rule prediction for the following reasons:
> >
> > - **Multiple Valid Solutions in RPMs:** RPM puzzles often have multiple valid solutions, meaning the model might generate solutions that differ from the given choices in the list. Comparing a generated solution with the choice list in the pixel space is therefore *not reliable*, as two solutions can look quite different while adhering to the same underlying rules. By focusing on the predicted rule matrix, we ensure a more robust approach to answer selection.
> >
> > - **Challenges with the "Generate and Compare" Approach:** While imagining and comparing a generated solution to the choices might seem more intuitive or human-like, this approach works well only for simple puzzles with unique solutions. For example, to address the issue with multiple valid solutions mentioned above, we need to sample a lot of generated answers, and compare all of them to the candidates. As puzzle complexity increases, the solution space grows severely, making it computationally expensive and impractical even for humans. Searching for a match in such a vast solution space would be time-consuming and frustrating.
> >
> > - **Human-Like Reasoning in Complex Scenarios:** In challenging puzzles, humans often rely on the choice list itself to identify the best solution by evaluating which option fits the puzzle's context most effectively. This strategy actually aligns with our inference algorithm, which selects the option that forms the largest number of rules, making it closer to how humans solve difficult RPM puzzles.
> >
> > For these reasons, we believe our rule-based inference algorithm is not only more efficient but also better reflects human reasoning in complex puzzle-solving scenarios.
> >
> > ### Response to Question 1
> > > In the graphical model of GenVP in Figure 1,
> >  is a shaded circle which means observation; however, from the context of model description it seems that
> >  is inferred.
> >
> > That is correct; in the inference graphical model, the rule matrix is inferred from the puzzle. However, for the generative graphical model, the rule matrix is marked as observed, allowing the process to start with a desired set of rules ($\mathbf{R}$) and generate a complete puzzle that adheres to these predefined (observed) rules.
> >
> > In summary, during inference, GenVP infers the rule matrix from the puzzle images without direct observation. Conversely, when generating novel puzzles, the rule matrix can be predefined and treated as observed.

---

> > > ### Author Response · Authors · 2024-11-23
> > > **Official Author Response (Part C)**
> > >
> > > ### Response to Question 2
> > > > Could the authors provide results on any other visual puzzles or abstract visual reasoning tasks? My main concern is whether GenVP can be applied to more realistic reasoning problems with a unspecific rule set.
> > >
> > > Thank you for mentioning this. Actually we did include experiments on five diverse datasets across different tasks:
> > >
> > > 1. **RAVEN**, **I-RAVEN**, and **RAVEN-FAIR**:
> > >    These datasets are widely used instantiations of Raven's Progressive Matrices (RPMs), which involve abstract reasoning tasks with structured rules.
> > >
> > > 2. **PGM (Procedurally Generated Matrices)**:
> > >    This dataset further evaluates RPM-solving capabilities for puzzles with increased solution spaces, including seven additional out-of-distribution training sets.
> > >
> > > 3. **VAD (Visual Analogy Dataset)**:
> > >    This dataset focuses on making visual analogies in a domain transfer setup, testing the generalization ability of models like GenVP to novel domains. VAD also has a larger solution space compared to RAVEN-based datasets.
> > >
> > >
> > > **Experiments with Unspecific Rule Sets:**
> > > Inspired by your comments, we conducted additional experiments to train GenVP without any rule annotations, using only the puzzle images and their correct candidate label. The results in **(Rebuttal) Table 2** (from the answer to Weakness 2) highlight GenVP's ability to perform reasonably well without explicit annotations, achieving significantly higher accuracy than prior work. This indicates that GenVP captures useful latent representations even without direct supervision of the rules. However, we recognize that there is room for improvement.
> > > In future work, we aim to explore self-supervised or weakly supervised approaches for rule discovery, which could significantly broaden the applicability of GenVP to unlabeled datasets. These directions will be highlighted in the revised manuscript.
> > >
> > > Overall, the datasets used in our experiments encompass a broad spectrum of abstract visual reasoning and analogy-making tasks. While the RPM datasets highlight GenVP’s strength in reasoning with structured rules, the VAD dataset showcases its ability to generalize to scenarios focused on visual analogy-making. With further refinement, GenVP holds great promise for addressing even more complex and realistic reasoning challenges in the future.

---

> > > > ### Author Response · Authors · 2024-11-25
> > > >
> > > > Dear Reviewer GBgR,
> > > >
> > > > Thank you for your constructive feedback and thoughtful questions about our paper. With the ICLR public discussion phase ending soon (Nov 26 AOE), we wanted to confirm whether our responses have sufficiently addressed your concerns. If there are any remaining issues, we would be happy to provide additional clarifications.
> > > >
> > > > We sincerely appreciate your feedback and the opportunity for discussion, and we kindly ask that you consider adjusting your evaluation if our responses have adequately resolved your concerns.
> > > >
> > > > Thank you so much for your time!

---

> > > > > ### Comment · Reviewer_GBgR · 2024-11-25
> > > > >
> > > > > Thanks for the detailed response, which has addressed most of my concerns. I still have some questions about the graphical model of GenVP. According to the response, GenVP seems to follow the paradigm of conditional generation, as the model generates an RPM corresponding to an observed rule matrix. As far as I know, in the inference process of VAE-based conditional generative models, the rule matrix should be an observed variable as well. However, the rule matrix is regarded as an inferred latent variable in GenVP, since it is derived from Z_o and Z_r in the posterior of the ELBO (Equation 7). Could the authors further clarify this point?

---

> ### Author Response · Authors · 2024-11-26
> **Rule Matrix in GenVP**
>
> Dear Reviewer,
>
> Thank you for your response. We are glad that our previous response addressed most of your concerns. Please let us know if there are any additional points we can help clarify.
>
> In fact, the rule matrix variable can be treated as a latent variable. In this case, our training objective is equivalent to the ELBO with a strong prior that forces the variational distribution of the rule latent variable $q_\phi(\mathbf{R}|\mathbf{Z}_o, \mathbf{Z}_r)$ to the ground-truth rules during training.
>
> $$L_R = \beta\_2\mathbb{E}\_{q}\left[ \log \frac{p(\mathbf{R})}{q\_{\boldsymbol{\phi}}(\mathbf{R}|\mathbf{Z}\_o,\mathbf{Z}\_{r})}\right] = \beta_2\left( \underbrace{\mathbb{E}\_{q}\left[ \log p(\mathbf{R}) \right]}\_{\text{Generative Graphical Model Term}} - \underbrace{\mathbb{E}\_{q}\left[ \log {q\_{\boldsymbol{\phi}}(\mathbf{R}|\mathbf{Z}\_o,\mathbf{Z}\_{r})}\right]}\_{\text{Inference Graphical Model Term}}\right) $$
>
> In practice, **during training**, the $L_R$ term reduces to a reconstruction loss between the ground truth rule matrix (which is observed in the generative graphical model) and the inference graphical model rule prediction ($q\_{\boldsymbol{\phi}}(\mathbf{R}|\mathbf{Z}\_o,\mathbf{Z}\_{r})$).
>
> After training GenVP, at test/inference time, we can consider separately the **generative graphical model** (depicted with red arrows in Figure 1) and the **inference graphical model** (depicted with black arrows in Figure 1).
>
> ---
>
> ### **Inference Graphical Model (Black arrows in Figure 1):**
>
> In the inference graphical model, a complete RPM puzzle $\mathbf{X}$ is observed. Following the black arrows in Figure 1, we infer/predict the rule latent variable $\mathbf{R}$. This prediction marks the end of the inference graphical model pipeline.
>
> ---
>
> ### **Generative Graphical Model (Red arrows in Figure 1):**
>
> In the generative graphical model, the rule matrix $\mathbf{R}$ serves as the "root" node. A user can select/observe a desired set of rules $\mathbf{R}$, and by following the red arrows in Figure 1, the model generates a complete puzzle $\mathbf{X}$ that adheres to the specified rules.
>
> For an observed/selected $\mathbf{R}$, the user has two options:
> 1. **Direct Rule Specification:** The user can design their own rule matrix $\mathbf{R}$ to generate puzzles.
> 2. **Derived Rule Specification:** The user can provide an existing puzzle $\mathbf{X}$, first pass it through the inference graphical model to predict $\mathbf{R}$, and then use this $\mathbf{R}$ as input to the generative graphical model to generate a new puzzle $\mathbf{X}'$. _In this case, we are using the inference graphical model because we start from a puzzle $\mathbf{X}$, but we need to emphasize that it is **not** necessary to use a puzzle $\mathbf{X}$ to generate a new puzzle $\mathbf{X}'$; someone can construct/observe a rule matrix $\mathbf{R}$ and construct a complete puzzle by just using the generative graphical model (learned decoders)._
>
> Both approaches are supported by GenVP, showcasing its flexibility.
>
> ---
>
> ### Summary
>
> In the **inference graphical model**, the rule matrix $\mathbf{R}$ is inferred from the given puzzle $\mathbf{X}$.
> In the **generative graphical model**, the rule matrix $\mathbf{R}$ is observed or specified, and it serves as the foundation for generating a new puzzle $\mathbf{X}$.
>
> Thus, the trained inference graphical model (learned encoders) predicts the rules $\mathbf{R}$ of a given puzzle $\mathbf{X}$, while the trained generative graphical model (learned decoders) generates a puzzle $\mathbf{X}'$ from a given set of rules $\mathbf{R}'$.

---

> ### Author Response · Authors · 2024-12-01
> **Examples of GenVP Applications**
>
> Dear Reviewer,
>
> Next, we provide examples outside the standard RPMs (like those found in RAVEN-based or PGM datasets) where GenVP can be employed, showcasing our approach's broader applicability.
>
> - **Flexibility with Input Format:**
>
>   GenVP is not limited to the $3 \times 3$ input format of RPM puzzles and can be readily generalized to arbitrary $M \times N$ puzzle settings with similarly arbitrary choice list size.
>
>
> - **Application to Question Answering (QA):**
>
>   More generally, the $M \times N$ puzzle settings can be interpreted as a form of Question Answering (QA) or dialogue, where the rows correspond to dialogue turns (e.g., $x_{i,1}, \ldots, x_{i,N-1}$ is the $i$-the question, $x_{i,N}$ is the answer to this question or the $i$-th turn dialogue completion). The puzzle can then be viewed as giving examples of $M-1$ QA pairs and seeking the answer $x_{M,N}$ to the last question $x_{M,1}, \ldots, x_{M,N-1}$, from the list of possible answers.
>
>   The key significance of applying our GenVP framework to the above setting is the ability to explicitly learn and interpret the attributes and rules that underpin this QA/dialogue and subsequently use them to generate new instances of QA/dialogue that employ these rules and/or attributes. This is in contrast to many other models today that approach similar QA/dialogue modeling problems in a completely agnostic, uninterpretable manner.
>
>
>   In visual puzzles, the dialogue elements $x_{i.j}$ are images, but it is not difficult to conceive that this could be generalized to combinations of images and words, leading to Visual Question Answering [(VQA)](https://visualqa.org/), or natural language QA alone.
>
>
> - **Application to Other Puzzle Settings:**
>
>   GenVP can also help solve Bongard problems, c.f., [1]. In Bongard-LOGO, we're given two groups of six images. One group follows rules A, and the other follows rules B. Our task is to match each of the two candidate images with the correct group based on shared properties. GenVP can be tailored for Bongard problems by replacing an image from each group with a candidate image. By evaluating which candidate image most consistently follows either set of rules, we identify the proper grouping for each candidate.
>
> [1] Nie, Weili, et al. "Bongard-logo: A new benchmark for human-level concept learning and reasoning." Advances in Neural Information Processing Systems 33 (2020): 16468-16480.

---

> ### Author Response · Authors · 2024-12-01
> **Kindly Reminder**
>
> Dear Reviewer GBgR,
>
> Thank you for your review and engagement during the discussion period.
>
> In response to your questions, we have provided explanations in our last two messages, specifically addressing the role of the rule matrix representation in GenVP (**Rule Matrix in GenVP** message) and presenting examples of applications beyond standard RPMs where GenVP can be utilized (**Examples of GenVP Applications** message).
>
> With the ICLR Discussion Period concluding soon (Dec. 2nd (AOE) for reviewers and Dec. 3rd (AOE) for authors), we kindly request your feedback on whether our responses address your concerns or if there are any additional questions or suggestions you would like us to address.
>
> Thank you once again for your time!
>
> Best,
>
> The Authors

---

> > ### Comment · Reviewer_GBgR · 2024-12-02
> >
> > Thanks for the further clarification. There are still some issues regarding modeling the rule matrix:
> > 1. According to the response, the rule matrix in GenVP is inferred from other latent variables during the inference process, so it should not be treated as observed variables in the graphical model (Figure 1).
> > 2. In auto-encoding variational Bayes, a random variable should not have different definitions in generation and inference processes. Generally, the observed variables in the generative process are also observed during inference, which is a precondition when approximating log-likelihood with ELBO for both conditional [1] and unconditional [2] modelling. However, the rule matrix is set observed in generation while regarded as a latent variable in inference. How can we guarantee the correctness of GenVP's ELBO?
> > [1] Kingma, Diederik P. "Auto-encoding variational bayes." arXiv preprint arXiv:1312.6114 (2013).
> > [2] Sohn, Kihyuk, Honglak Lee, and Xinchen Yan. "Learning structured output representation using deep conditional generative models." Advances in neural information processing systems 28 (2015).

---

> ### Author Response · Authors · 2024-12-02
>
> Dear Reviewer,
>
> We apologize for the confusion; this is an important point.
>
> Firstly, let us consider the standard VAE model,
>
> $\mathbb{E}\_{q(\mathbf{z}|\mathbf{x})}\left[\log p(\mathbf{x}|\mathbf{z})\right] - KL(q(\mathbf{z}|\mathbf{x}) || p(\mathbf{z}))$
>
> in which case we have the KL term:
>
> $KL(q(\mathbf{z}|\mathbf{x}) || p(\mathbf{z}))$
>
> Then, for the prior distribution of $p(\mathbf{z})$, we can assume it follows a prior distribution $\mathcal{N}(0, \epsilon)$, with $\epsilon \to 0$. Consider $\mathbf{z}$ like our rule matrix.
>
> Similarly, in GenVP, we model the rule matrix categorical prior ($p(\mathbf{R})$) parameters with Dirichlet priors with very large values for the concentration hyperparameters $\alpha$.
>
> The above modeling ensures our GenVP's ELBO correctness. We have included this explanation in our manuscript as well. Thank you for pointing out this important point, which helped us improve our paper's clarity.

---

> > ### Comment · Reviewer_GBgR · 2024-12-02
> >
> > Thank the authors for the detailed response. I really appreciate the authors' effort to clarify the issues about the ELBO. I'm not sure if the authors fully understood my concerns. Let’s take the standard VAE as an example. In the ELBO, the prior $p(\boldsymbol{z})$ indicates the generative process, and the variational posterior $q(\boldsymbol{z}|\boldsymbol{x})$ indicates the inference process, where $\boldsymbol{x}$ is the set of observed variables and $\boldsymbol{z}$ is the set of latent variables. In GenVP, $\boldsymbol{R}$ is observed in the generative process, thus we have $\boldsymbol{R} \in \boldsymbol{x}$ for the prior $p(\boldsymbol{x})$. But during the inference process, $\boldsymbol{R}$ is an inferred latent variable, indicating that $\boldsymbol{R} \in \boldsymbol{z}$ for $q(\boldsymbol{z}|\boldsymbol{x})$. My concern is that the way GenVP defines $\boldsymbol{R}$ may violate some assumptions used in deriving the ELBO of standard VAE, it is unclear that $\boldsymbol{R}$ is a latent variable or an observed variable, hence the correctness of GenVP's ELBO is still unclear.

---

> > > ### Author Response · Authors · 2024-12-02
> > >
> > > Dear Reviewer,
> > >
> > > Thank you for your follow-up and for helping us better understand your point.
> > >
> > > **$\mathbf{R}$ as a Latent Variable.** You are correct; in GenVP, the rule matrix $\mathbf{R}$ is indeed a latent variable.
> > >
> > > However, during training, we obtain the data { $(X_i, R_i)$ } $_{i=1}^N$, where $N$ is the total number of annotations, and we want to use the rule annotations to reduce our uncertainty over the latent variable $\mathbf{R}$. _We will use separate notation for the latent variable $\mathbf{R}$ and the observation $\mathbf{R}\_i$._
> > >
> > > **Validity of ELBO.** Returning to our ELBO formulation, the second term of Eq 7:
> > >
> > > $$ \mathcal{L}\_{R} = \beta\_2\mathbb{E}\_{q}\left[ \log \frac{p(\mathbf{R})}{q\_{\boldsymbol{\phi}}(\mathbf{R}|\mathbf{Z}\_o,\mathbf{Z}\_{r})}\right]$$
> > >
> > > is the KL divergence between our variational approximate posterior $q_{\boldsymbol{\phi}}(\mathbf{R}|\mathbf{Z}_o, \mathbf{Z}_r)$ and our special (highly concentrated) prior $p(\mathbf{R})$.
> > >
> > > During training (Eq 12), for the i-th data point $(\mathbf{X}_i, \mathbf{R}_i)$, where $\mathbf{X}_i$ is the input image and $\mathbf{R}_i$ is the rule. We will use a prior $p(\mathbf{R}|\mathbf{R}_i)$ that is highly concentrated around $\mathbf{R}_i$ and a variational approximate posterior $q(\mathbf{R}|\mathbf{Z}_o, \mathbf{Z}_r)$. Note that differently from standard VAE, our prior is data-dependent, i.e., depending on $\mathbf{R}_i$. In this case, $q(\mathbf{R}|\mathbf{Z}_o, \mathbf{Z}_r)$ is a probability distribution, and $p(\mathbf{R}|\mathbf{R}_i)$ is also a probability distribution, but it is approaching a Dirac Delta distribution $p(\mathbf{R}|\mathbf{R}_i) = \delta(\mathbf{R}-\mathbf{R}_i)$, see https://en.wikipedia.org/wiki/Dirac_delta_function.
> > >
> > > Therefore, the ELBO is still valid. Intuitively, minimizing this term is equivalent to training a supervised learning loss, i.e., training a rule predictor to predict $\mathbf{R}$ given $\mathbf{Z}_0$ and $\mathbf{Z}_r$.
> > >
> > > **Similar Formulation in Prior Work [1].** We would also like to mention that a similar problem has been explored in prior work [1], where the additional observations are used through the introduction of auxiliary distributions (see Eqs 7-9 in [1]) resembling our data-dependent rule matrix prior. The available observations are used in the last term of Eq. (10) in [1] to estimate the parameters of the auxiliary distributions. The second term of Eq. (10), is similar to our treatment of the rule annotation $\mathbf{R}_i$.
> > >
> > > We hope the above explanation addresses your concerns. We have updated our manuscript, and in Figure 1, we separated the latent variable $\mathbf{R}$ from the observation $\mathbf{R}_{i}$, which we use during training to reduce our uncertainty about the rule matrix latent variable.
> > >
> > > Thank you once again for your efforts to improve our manuscript.
> > >
> > > [1] Louizos, C., Shalit, U., Mooij, J. M., Sontag, D., Zemel, R., & Welling, M. (2017). Causal effect inference with deep latent-variable models. Advances in neural information processing systems, 30.

---

> > > > ### Comment · Reviewer_GBgR · 2024-12-03
> > > >
> > > > Thanks for the author's response. I have read the reference paper as well as the authors' reply. If my understanding is correct, the authors attempt to supervise GenVP with the rule annotations, making the definition of rule matrix unclear in the initial version of manuscript. I think the reference paper is a good example, as it uses additional log-likelihood losses to supervise the variables with labels. Maybe the authors can similarly formulate GenVP as a conditional generative model p(X|R), and supervise the rule matrix via \log q(R'|Z). Such training objective will not violate the derivation of standard VAE's ELBO.
> > > >
> > > >
> > > > The authors clarify that the rule matrix is actually a latent variable and promise to modify the graphical model in the revised version. I believe this will make the definition of GenVP clear. The authors try to introduce a new observed variable R' to explain the ELBO, and they mention that p(R) is a data-dependent prior. However, these definitions are not mentioned by the authors in the previous manuscripts or discussions. In my opinion, this is a **correction to the original version** of GenVP rather than a clarification of its correctness. I am glad to see that, after rounds of discussions during rebuttal, the new way has made the definition of GenVP correct and clearer. I believe the motivation of discovering the underlying structure to generate visual puzzles is commendable. I encourage the authors to correctly define the model to improve the paper in the next version.

---

> ### Author Response · Authors · 2024-12-04
>
> Dear Reviewer GBgR,
>
> Thank you for your response. We are glad you found the data-dependent prior formulation helpful for illustrating how the rule annotations were used during training to learn robust rule predictions in GenVP. We would also like to thank you for recognizing the importance of our work.
>
> We want to reiterate that our latest response on the data-dependent prior is **consistent** with our original paper. In Eq. 7 of the original paper, $p(\mathbf{R})$ is the prior distribution of a latent variable $\mathbf{R}$; therefore, the only (minor) change needed is to clarify that “$p(\mathbf{R})$ is data-dependent prior” in the revision. This **clarification is certainly important and will be included in the camera-ready version**.
>
> Finally, we kindly ask if you might consider adjusting your rating if you agree that our response has effectively addressed your concerns and added value to the understanding of our work.
>
> Thank you for your time and constructive feedback.
>
> Best,
>
> The Authors

---

### Official Review · Reviewer_YfAj · 2024-11-02

**Soundness:** 3
**Presentation:** 4
**Contribution:** 2
**Rating:** 6
**Confidence:** 5

**Summary:**

The authors proposed a novel deep latent variable model GenVP for RPM problems. GenVP extracts image features and then uses these features to form relevant and irrelevant representations for later inference. The proposed MoE estimator can infer categories of attribute-level rules used to generate RPM panels. The proposed model was evaluated through experiments on RAVEN/I-RAVEN, VAD and PGM, showing better performance than other generative-based methods.

**Strengths:**

1.	This paper is generally well-written and the main idea is easy to follow. The authors proposed a new generative model GenVP to encode rule-related information. It will be more useful to construct interpretable machine learning algorithms for abstract visual reasoning.
2.	Compared to the previous generative approach RAISE, GenVP is unconditional and therefore can generate novel RPMs without given context.
3.	The decomposition of relevant and irrelevant representations improves the robustness of GenVP when there is too much noise or resolution in puzzles.
4.	The authors showed that their GenVP can perform better than previous generative methods.

**Weaknesses:**

1.	Many important factors in GenVP should be carefully tuned. For example, and in the sampling process should be carefully handcrafted.
2.	The proposed model seems to rely heavily on ground truth rule annotations. GenVP leverages rule annotations of each RPM sample in the training process. This limits the applicability of this method to unlabelled visual reasoning datasets. The authors should investigate why the models cannot learn the rules well without annotations.
3.	Using VAE-based generative solvers or contrastive loss is not novel in RPMs. Is there any novel technical or insightful design in GenVP compared to previous approaches.

**Questions:**

1.	Could the authors explain the process of answer selection in detail?
2.	Will the selection of hyperparameters heavily influence the performance of GenVP? The RPM datasets contain just small sets of object attributes and rules, and the selection of hyperparameters can probably matter a lot in this case. The authors should discuss this as well.

---

> ### Author Response · Authors · 2024-11-23
> **Official Author Response (Part A)**
>
> ### Response to Weakness 1
> > Many important factors in GenVP should be carefully tuned. For example, and in the sampling process should be carefully handcrafted.
>
> We appreciate the reviewer’s concerns regarding factor tuning and the sampling process in GenVP. We would like to clarify that GenVP’s strong performance is **not** due to excessive fine-tuning but rather to its sophisticated design and integration of generative and contrastive training schemes.
>
> - **Robustness Across Datasets and Architectures:** GenVP achieves state-of-the-art results across five datasets (RAVEN, I-RAVEN, RAVEN-FAIR, PGM, and VAD), including challenging ones like PGM and VAD, where existing methods, such as RAISE, struggle to generalize. Importantly, these results were achieved without extensive re-tuning across different architectures (CNN, ResNet, ViT) or image resolutions (64x64, 128x128, 224x224), as demonstrated in our ablation study (Table 13, Section D.2).
>
> - **Minimal Hyperparameter Adjustments:** The only notable adjustment was for the rule matrix ELBO term $L_R$. In particular for $L_R$, $\beta_2$ was increased to balance the dimensions of the rule latent variables and the pixel-space image vectors. This adjustment is straightforward and ensures effective optimization without requiring extensive experimentation.
>
> - **Sampling Process:** The sampling process in GenVP follows a standard VAE-based framework. Each intermediate latent variable is modeled as a Gaussian random variable, with GenVP predicting its mean and variance for straightforward sampling. Thus, the process is *not handcrafted* but *aligns with established probabilistic modeling practices*.
>
> Overall, the consistent results across datasets, architectures, and resolutions underscore GenVP’s robustness and adaptability, which we attribute to its *principled design rather than meticulous fine-tuning*. We have included the discussion above in our revised paper.  We also performed a number of new experiments for different hyperparameter values (See our response to **Question 2** ( experiment results in **Rebuttal Tables 3 and 4**)).
>
> ### Response to Weakness 2
> > The proposed model seems to rely heavily on ground truth rule annotations. GenVP leverages rule annotations of each RPM sample in the training process. This limits the applicability of this method to unlabelled visual reasoning datasets. The authors should investigate why the models cannot learn the rules well without annotations.
>
> Thank you for your question. We would like to clarify that all existing models addressing RPM tasks (GCA, ALANS, PRAE, RAISE) also rely on these rule annotations. Therefore our comparison is fair.
>
> We also agree that designing models capable of abstract reasoning without relying on rule annotations is indeed an essential challenge for advancing unsupervised visual reasoning.
>
> GenVP benefits from rule annotations to learn structured, interpretable representations, ensuring that the generated puzzles align with specific rules. This approach provides a level of control and explainability not seen in other methods.
> However, we acknowledge the desire for methods that minimize supervision and can generalize to unannotated datasets.
>
> **New Results without Rule Annotation:** Inspired by your comments, we conducted additional experiments to train GenVP without any rule annotations, using only the puzzle images and their correct candidate label. The results in **(Rebuttal) Table 2** highlight GenVP's ability to perform reasonably well without explicit annotations, achieving significantly higher accuracy than prior work. In particular, for RAVEN, GenVP improves the performance of the prior work, RAISE, from 54.5 to 70.3 by $\approx 16$ points. This indicates that GenVP captures useful latent representations even without direct supervision of the rules. However, we recognize that there is room for improvement.
> In future work, we aim to explore self-supervised or weakly supervised approaches for rule discovery, which could significantly broaden the applicability of GenVP to unlabeled datasets. These directions will be highlighted in the revised manuscript.
>
> _**(Rebuttal) Table 2**: GenVP performance when trained without rule annotations._
>
> | Dataset                     | RAVEN | I-RAVEN |
> |-----------------------------|-------|---------|
> | RAISE (no rule annotations) | 54.5  | 67.7    |
> | GenVP (no rule annotations) | 70.3  | 73.8    |
>
> We appreciate the reviewer’s suggestions and hope that these results demonstrate GenVP's potential to generalize beyond annotated settings and improve on this compared to the previous SOTA model RAISE (ICLR 2024).

---

> ### Author Response · Authors · 2024-11-23
> **Official Author Response (Part B)**
>
> ### Response to Weakness 3
> > Using VAE-based generative solvers or contrastive loss is not novel in RPMs. Is there any novel technical or insightful design in GenVP compared to previous approaches.
>
> Thank you for your question. Since the same question was posed by Reviewer YBT2 in Weakness 1, to avoid duplicate answers, we kindly ask you to see our response to Reviewer YBT2 (above).
>
> ### Response to Question 1
> > Could the authors explain the process of answer selection in detail?
>
> To select the correct or most likely answer in an RPM puzzle, we rely on the properties of the most likely solution, which is the candidate that forms the largest number of rules when placed in the context matrix. Next we describe the steps for finding the puzzle solution from a pre-trained GenVP model.
>
> **Context Matrix Completion**: For each answer option in the choice list, we create eight (i.e. in RAVEN-based, PGM datasets) complete candidate puzzles by filling the missing bottom-right position in the context matrix with one of the answer choices. (Each complete candidate puzzle now has 9 images.)
>
> **Rule Prediction with GenVP**: For each complete candidate puzzle, we apply the GenVP inference pipeline to predict the puzzle rules (using the Mixture of Experts (MoE)).
>
> **Answer Selection**: We then select the candidate that maximizes the number of active rules in the rule matrix prediction, including rules like constant, progression, distribute-three, and arithmetic. This is the same as selecting the candidate minimizing the number of random/no-rule in the rule matrix prediction.
> In practice, we return the answer choice that minimizes the count of random/no-rule encodings in the matrix (for example, the encoding $10000$ indicates a no-rule or random pattern, so we aim to minimize occurrences of this encoding).

---

> ### Author Response · Authors · 2024-11-23
> **Official Author Response (Part C)**
>
> ### Response to Question 2 (Part A: Optimization Objective Hyperparameters and GenVP Performance)
> > Will the selection of hyperparameters heavily influence the performance of GenVP? The RPM datasets contain just small sets of object attributes and rules, and the selection of hyperparameters can probably matter a lot in this case. The authors should discuss this as well.
>
> This is a good question. Following the reviewer's suggestion, we conducted additional ablation studies to evaluate the influence of hyperparameters on GenVP's performance. Our results, shown in **(Rebuttal) Table 3**, indicate that GenVP is not overly sensitive to hyperparameter tuning, as performance remains robust across a range of configurations.
>
> **(Rebuttal) Table 3:** _GenVP Puzzle Solving Accuracy for the RAVEN dataset. We train four different versions, using different hyperparameter values $\beta_i, \beta_G, \beta_L$ for $i=1,\dots,6$._
> | $\beta_1$ | $\beta_2$ | $\beta_3$ | $\beta_4$ | $\beta_5$ | $\beta_6$ | $\beta_G$ | $\beta_L$ | Accuracy |
> |-----------|-----------|-----------|-----------|-----------|-----------|-----------|-----------|-------------------------|
> | 1         | 250       | 1         | 1         | 1         | 1         | 20        | 20        | 94.7                    |
> | 1         | 1         | 1         | 1         | 1         | 1         | 20        | 20        | 95.0                    |
> | 1         | 1         | 1         | 1         | 1         | 1         | 1         | 1         | 92.1                    |
> | 1         | 1         | 1         | 1         | 1         | 1         | 40        | 40        | 94.9                    |
>
> Below we discuss these new results in detail.
>
> 1. **Rule Matrix Weight ($\beta_2$)**:
>    In our primary experiments, we assigned a higher weight ($\beta_2 = 250$) to the rule matrix prediction term compared to other ELBO terms ($\beta_i = 1, i \neq 2 $). To test sensitivity, we set all $ \beta_i, i=1,2,\dots,6 $ to 1, effectively removing the emphasis on the rule matrix. This resulted in an improvement in performance (~94.7% to ~95.0%) demonstrating that the model is robust even when the weighting is altered.
>
> 2. **Contrastive Loss Weights ($ \beta_G, \beta_L $)**:
>    In our main experiments, $ \beta_G $ and $ \beta_L $ were set to 20. In the additional experiments, we evaluated three configurations ($ \beta_G, \beta_L = 1, 20, 40 $). Results showed that very small weights ($ \beta_G, \beta_L = 1 $) caused a modest degradation (~92.1% accuracy), but increasing the weights ($ \beta_G, \beta_L = 20, 40 $) stabilized performance at ~95%. This suggests that GenVP benefits from reasonable tuning of contrastive terms but does not require fine-grained optimization.
>
> 3. **Performance Consistency Across Hyperparameters**:
>    Across all tested hyperparameter configurations, GenVP maintained strong puzzle-solving accuracy. The observed variations in performance were within an acceptable range, indicating that the model’s core design, alongside its generative and contrastive training schemes is robust to hyperparameter changes.
>
> In conclusion, while thoughtful adjustment of certain hyperparameters, such as $ \beta_2 $, $ \beta_G $, and $ \beta_L $, can optimize performance, GenVP does not rely heavily on precise tuning. Its stability across a wide range of settings underscores its adaptability and reliability in RPM puzzle solving tasks.
>
> (continued to Question 2 Part B...)

---

> > ### Author Response · Authors · 2024-11-25
> >
> > Dear Reviewer YfAj,
> >
> > Thank you for your constructive feedback and thoughtful questions about our paper. With the ICLR public discussion phase ending soon (Nov 26 AOE), we wanted to confirm whether our responses have sufficiently addressed your concerns. If there are any remaining issues, we would be happy to provide additional clarifications.
> >
> > We sincerely appreciate your feedback and the opportunity for discussion, and we kindly ask that you consider adjusting your evaluation if our responses have adequately resolved your concerns.
> >
> > Thank you so much for your time!

---

> ### Comment · Reviewer_YfAj · 2024-11-25
>
> I appreciate the authors for the detailed response. Most of my questions have been resolved. The additional results illustrate the efficiency of tuning hyperparameters related to the training ELBO. But I still wonder if GenVP is sensitive to the hyperparameters of the model architecture, e.g., $K$, $K_{Z_o}$, $K_R$, $N_R$? I noticed that these hyperparameters are differently configured across datasets.

---

> ### Author Response · Authors · 2024-11-28
> **Official Author Response (Part D)**
>
> ### Response to Question 2 (Part B: Hyperparameters for Latent Space Dimensionality and GenVP Performance)
>
> Dear Reviewer YfAj,
>
> We conducted additional experiments to explore the impact of latent variable size on GenVP's performance.
>
> **(Rebuttal) Table 4:** _GenVP Puzzle Solving Accuracy for the RAVEN dataset. We trained five different versions of the model, using varying hyperparameter values $K$, $K\_{Z\_o}$, $K\_{Z\_{\bar{o}}}$, $K\_{Z\_r}$, $K_R$, and $N_R$, which control the size of GenVP latent variables._
>
> | GenVP Total Number of Parameters | $K$  | $K_{Z_o}$ | $K_{Z_{\bar{o}}}$ | $K_{Z_r}$ | $K_R$ | $N_R$ | Accuracy |
> |-----------------------|------|-----------|--------------------|-----------|-------|-------|----------|
> | 8,250,313            | 32   | 20        | 12                 | 60        | 12    | 5     | 91.6     |
> | 8,912,077            | 64   | 54        | 10                 | 162       | 22    | 5     | 95.3     |
> | 8,957,157            | 64   | 54        | 10                 | 162       | 12    | 15    | 92.4     |
> | 9,758,185            | 128  | 100       | 28                 | 300       | 12    | 5     | 94.8     |
> | 10,208,245           | 128  | 126       | 2                  | 378       | 12    | 5     | 94.9     |
>
> As observed in **(Rebuttal) Table 4**, GenVP consistently achieves high accuracy across a range of hyperparameter configurations.
>
> Notably, if the latent variable dimensions and total number of model parameters are too restrictive (do not have enough capacity to capture the puzzle complexity), we expect GenVP's performance to be affected. However, although representing each pixel-level image of RAVEN with a latent vector $\mathbf{z}_{ij}$, $i,j\in\{1,2,3\}$ of dimensionality $K=32$ is restrictive (prior work, i.e. RAISE, uses 64-dimensional image encodings in the latent space), GenVP manages to achieve an accuracy of 91.6\% (first row in **(Rebuttal) Table 4**). As the model total parameters and the capacity of the latent space vectors increase, GenVP's accuracy improves and stabilizes at $\sim95$\%. This indicates that GenVP is robust to architectural hyperparameter choices, though models with large enough latent vectors generally yield better results.

---

> > ### Author Response · Authors · 2024-12-01
> > **Kindly Reminder**
> >
> > Dear Reviewer YfAj,
> >
> > Thank you for your review and engagement during the discussion period.
> >
> > In response to your suggestions, we conducted additional ablation studies to analyze GenVP's performance across varying sizes of latent variables, as shown in **(Rebuttal) Table 4**. These results, combined with those in **(Rebuttal) Table 3**, demonstrate that GenVP achieves consistently high performance across various hyperparameter configurations.
> >
> > With the ICLR Discussion Period concluding soon (Dec. 2nd (AOE) for reviewers and Dec. 3rd (AOE) for authors), we kindly request your feedback on whether our responses address your concerns or if there are additional questions or suggestions you would like us to address.
> >
> > Thank you once again for your time!
> >
> > Best,
> >
> > The Authors

---

> ### Comment · Reviewer_YfAj · 2024-12-01
>
> Thanks for the additional experiment on tuning architecture-related hyperparameters. The results provided by the authors have addressed my concerns. I will raise score to 6.

---

### Official Review · Reviewer_YBT2 · 2024-11-05

**Soundness:** 3
**Presentation:** 2
**Contribution:** 2
**Rating:** 6
**Confidence:** 4

**Summary:**

Briefly, this paper presents a framework to solve and create complete new puzzles out of the desired set of rules for Raven’s Progressive Matrices (RPM) by introducing the contrastive learning scheme (i.e., cross-puzzle and cross-candidate contrastive loss) and MoE mechanism for puzzle rule prediction. The experimental results are strong.

**Strengths:**

+ The paper is well-written and easy to follow. The idea of the proposed method is quite interesting.
+ Extensive and comprehensive experiments demonstrate the effectiveness of the proposed method.

**Weaknesses:**

1) The technical novelty of the proposed method seems to be marginal since the authors directly employ the existing techniques (e.g., contrastive learning scheme and MoE). More detailed discussions and analyses are required to demonstrate the contribution of the proposed method.

2) More ablation studies are required to demonstrate the contribution of the main component of the proposed method in the main paper. For instance, the mentioned rule estimators for the RPM-Level inference, the performance gain of the introduced MoE mechanism, and the contrastive learning scheme.

**Questions:**

Please refer to the Weaknesses section.

---

> ### Author Response · Authors · 2024-11-23
> **Official Author Response (Part A)**
>
> ### Response to Weakness 1
> > The technical novelty of the proposed method seems to be marginal since the authors directly employ the existing techniques (e.g., contrastive learning scheme and MoE). More detailed discussions and analyses are required to demonstrate the contribution of the proposed method.
>
> Thank you for your question. Below, we would like to clarify the novel technical and insightful design choices in GenVP compared to prior approaches:
>
> #### A Novel Task - Complete RPM Puzzle Generation:
>
> - GenVP is the first VAE-based model to tackle the complete RPM puzzle generation process. Previous works focused solely on puzzle solution generation, leaving the broader task of complete puzzle modeling unexplored.
> - This shift in scope required the development of a novel graphical model enabling controlled and interpretable generation, which the proposed model, GenVP, embodies.
>
> #### Novel Disentangled Latent Representations:
>
> - GenVP introduces a disentangled latent space design, separating rule-relevant ($\mathbf{Z}\_o$) and rule-irrelevant ($\mathbf{Z}_{\bar{o}}$) factors. This data-driven disentanglement is both *novel* in RPMs and *crucial* for achieving robustness in puzzle generation and solving.
>
> #### Novel Mixture of Experts (MoE) Design:
>
> - The MoE architecture in GenVP is a *sophisticated* addition that enables precise and modular rule prediction. It avoids shortcut learning by leveraging both fully (representations from all puzzle images, i.e., $\mathbf{Z}_r, \mathbf{Z}_o$) and partially observed (a subset of puzzle images like $\mathbf{Z}^{prow}_o, \mathbf{Z}^{ctx}_o$) information across coarse ($\mathbf{Z}_r$) and fine ($\mathbf{Z}_o$) representations.
> - This design is inspired by human reasoning: focusing on various local (e.g., combinations of two rows of a puzzle) and global (complete puzzle) relations to infer rules effectively.
>
> #### Novel Contrastive Learning Scheme:
>
> - GenVP’s contrastive scheme is novel and different from typical contrastive learning in two key ways:
>   - **Complete Puzzle Comparison:** Instead of comparing single images like typical contrastive learning, GenVP compares entire puzzles, capturing global contextual dependencies.
>   - **High-Level Rule Representations:** Unlike typical contrastive learning's low-level pixel or latent comparisons, GenVP operates on high-level rule representations. This ensures the model focuses on abstract reasoning rather than superficial visual similarities.
> - Furthermore, GenVP’s contrastive learning contrasts valid and invalid puzzles, as well as different valid puzzles, offering a richer training signal that improves both generalization and interoperability.
>
> #### Novelty in Visual Puzzle Reasoning Process:
>
> GenVP focuses independently on different sub-parts of the puzzle to extract information about the relations and rules formed within those subsets. Ultimately, GenVP analyzes puzzles both locally (puzzle subsets, such as two rows) and globally (the complete puzzle). This approach mimics a meticulous reasoning process, carefully examining the given problem (RPM) to uncover the rules governing the visual puzzles.
>
> -------
> In summary, GenVP is not merely an incremental improvement but a significant conceptual and technical leap in generative RPM solvers. Its novel disentangled representations, advanced MoE design, and innovative contrastive learning scheme distinguish it clearly from prior approaches, both in generative capabilities and interpretative reasoning. These contributions are directly reflected in GenVP’s superior performance across diverse datasets and its ability to model the complete puzzle generation process.

---

> ### Author Response · Authors · 2024-11-23
> **Official Author Response (Part B)**
>
> ### Response to Weakness 2
> > More ablation studies are required to demonstrate the contribution of the main component of the proposed method in the main paper. For instance, the mentioned rule estimators for the RPM-Level inference, the performance gain of the introduced MoE mechanism, and the contrastive learning scheme.
>
> We thank the reviewer for their ablation study suggestions.
> In the Appendix of our submission, we included the following ablation studies to demonstrate the contribution of the main components of the proposed method:
>
> 1. **Contrastive Learning Effect**
>    **Tables 8 and 9 in the Appendix** present GenVP's performance without the contrastive learning scheme, using only ELBO maximization (Equation 7). Our experimental results demonstrate the critical role of contrastive learning in enabling GenVP to detect puzzles with the strongest visual analogies—those satisfying the largest number of rules. This establishes the importance of the contrastive learning scheme for the model's puzzle-solving capabilities.
>
> 2. **Effect of Different Ensembling Functions in the MoE Module**
>    We present different types of MoE function in **Section C.6**. and the corresponding performance in **Tables 10 and 11**.
>
> **Performance of Individual Rule Predictors vs. Mixture of Experts (MoE)**
> Following our reviewer's suggestions, we compared GenVP’s performance using individual rule predictors against its performance using the MoE (all predictors) for the RAVEN dataset. The results are summarized in Table 1.
>
> _**(Rebuttal) Table 1**: In the MoE column, we present the puzzle-solving accuracy using all rule predictors. In the remaining columns, we show the accuracy when using individual rule predictors $R(Z)$, where $Z$ denotes the input used for the individual rule prediction. For notation simplicity in the Table, for the rule predictions using only two rows, we use the notation $\mathbf{Z}^{prow}\_{ij}$ meaning that we use the two rows $i$ and $j$, for $i,j=1,2,3$. For the context-based rule predictions (one image is missing from the puzzle), we use the notation $\mathbf{Z}^{ctx}\_{-i}$ meaning that the $i$-th image of the puzzle is missing for $i=1,\dots,9$._
>
> | MoE  | $R(\mathbf{Z}_o)$ | $R(\mathbf{Z}_r)$ | $R(\mathbf{Z}_{12}^{prow})$ | $R(\mathbf{Z}^{prow}_{13})$ | $R(\mathbf{Z}^{prow}_{23})$ | $R(\mathbf{Z}^{ctx}_{-1})$ | $R(\mathbf{Z}^{ctx}_{-2})$ | $R(\mathbf{Z}^{ctx}_{-3})$ | $R(\mathbf{Z}^{ctx}_{-4})$ | $R(\mathbf{Z}^{ctx}_{-5})$ | $R(\mathbf{Z}^{ctx}_{-6})$ | $R(\mathbf{Z}^{ctx}_{-7})$ | $R(\mathbf{Z}^{ctx}_{-8})$ | $R(\mathbf{Z}^{ctx}_{-9})$ |
> |------|------------------|------------------|--------------------|--------------------|--------------------|--------------------|--------------------|--------------------|--------------------|--------------------|--------------------|--------------------|--------------------|--------------------|
> | 95.0 | 48.7             | 93.2             | 12.4               | 87.0               | 87.1               | 81.7               | 82.8               | 80.6               | 82.0               | 82.9               | 81.7               | 16.3               | 16.7               | 12.8               |
>
> (continued in next message)

---

> ### Author Response · Authors · 2024-11-28
> **Official Author Response (Part C)**
>
> ### Response to Weakness 2 (continued)
>
> In **(Rebuttal) Table 1**, we observe the following:
> - **State-of-the-Art Performance with MoE**: GenVP achieves the best performance (95.0% accuracy) when combining rule predictors through the MoE mechanism, demonstrating its effectiveness.
> - **Performance of Individual Predictors**: Among individual predictors, the puzzle-level representation predictor $R(\mathbf{Z}\_r)$ achieves the highest performance, followed by predictors using two puzzle rows ($R(\mathbf{Z}\_{13}^{prow})$, $R(\mathbf{Z}\_{23}^{prow})$) and context-based predictors ($R(\mathbf{Z}^{ctx}\_{-1})$, $R(\mathbf{Z}^{ctx}\_{-2})$, ..., $R(\mathbf{Z}^{ctx}\_{-6})$). Predictors using image-level representations ($R(\mathbf{Z}\_o)$) show comparatively lower performance.
>
>   The high performance of $R(\mathbf{Z}_r)$ indicates that the puzzle-level latent variables exclude most of the information irrelevant to the rules (noise), making them robust for the puzzle-solving task.
>
>   Additionally, we observe that the rule predictors focusing on only two rows outperform $R(\mathbf{Z}_o)$, which observes all three rows. By restricting these predictors to derive the rules from just two rows instead of three, the problem becomes more challenging, leading to greater robustness. This restriction prevents these predictors from exploiting shortcuts for rule prediction. For example, in the case of $R(\mathbf{Z}_o)$, the model might predict rules by focusing only on the first two rows of the puzzle, which would fail in scenarios where the third row contains an imposter image.
> - **Challenging Scenarios**: The context predictors ($R(\mathbf{Z}^{ctx}\_{-7})$, $R(\mathbf{Z}^{ctx}\_{-8})$) perform poorly when the missing context matrix image lies in the same row as the negative candidate. This occurs because, in such cases, even though the last row contains an imposter image, the absence of the 7th or 8th image allows GenVP to imagine a plausible missing image that aligns with the puzzle rule.
>
>   For example, consider a puzzle governed by an arithmetic rule based on the number of objects. The correct puzzle might have a number of objects (1,2,3; 2,2,4; 3,2,5), where the third image in each row is the sum of the first two. Now, suppose the last row contains an incorrect image, such as (1,2,3; 2,2,4; 3,2,4). This row violates the arithmetic rule, and GenVP should identify this as a puzzle with an incorrect candidate.
>
>   However, if we consider the $R(\mathbf{Z}^{ctx}_{-7})$ predictor and observe a partially complete matrix, such as (1,2,3; 2,2,4; 3,-,4), this puzzle could still be valid under the arithmetic rule if the missing image were (1,2,3; 2,2,4; 3,**1**,4). Thus, when the 7th or 8th image is removed from a puzzle containing an imposter image in the 9th panel, GenVP can imagine a plausible image that corrects the rule violation, leading to a performance drop for these context predictors.
> - **Random Performance**: Predictors like $R(\mathbf{Z}\_{12}^{prow})$ and $R(\mathbf{Z}^{ctx}\_{-9})$ exhibit random performance because the choice list images are missing, leading to identical images across all candidate puzzles. This result is both expected and desired, as the subsets of the candidate puzzles are identical in the image space and in GenVP's latent space, resulting in random performance.
>
>
> These ablations provide insights into the contributions of the contrastive learning scheme, MoE mechanism, and rule estimators to GenVP’s superior performance. We have included the MoE vs. individual rule predictors' performance in the main paper.

---

> > ### Author Response · Authors · 2024-11-28
> >
> > Dear Reviewer YBT2,
> >
> > Thank you for your constructive feedback and thoughtful questions about our paper. With the ICLR public discussion phase ending soon (December 2 AOE), we wanted to confirm whether our responses have sufficiently addressed your concerns. If there are any remaining issues, we would be happy to provide additional clarifications.
> >
> > We sincerely appreciate your feedback and the opportunity for discussion, and we kindly ask that you consider adjusting your evaluation if our responses have adequately resolved your concerns.
> >
> > Thank you so much for your time!

---

> ### Author Response · Authors · 2024-12-01
> **Kindly Reminder**
>
> Dear Reviewer YBT2,
>
> Thank you for your review. We would like to provide a summary of our posted rebuttal:
>
> + In our **Official Author Response (Part A)** message, we clarified the novel technical contributions and insightful design choices underlying GenVP.
> + In **Official Author Response (Part B)**, we referred to the ablation studies provided in our supplementary material and conducted additional ablations as requested. These new ablations investigate the role and performance of each rule predictor in GenVP, offering further insights into their individual contributions.
> + In **Official Author Response (Part C)**, we provided a detailed discussion on the MoE components in GenVP, elaborating on their role and effectiveness.
>
> With the ICLR Discussion Period ending soon (Dec. 2nd (AOE) for reviewers and Dec. 3rd for authors (AOE)), we would like to kindly ask for your feedback on whether our responses address your concerns or if you have additional questions or suggestions for clarification.
>
> Thank you again for your time!
>
> Best,
>
> The Authors

---

> > ### Comment · Reviewer_YBT2 · 2024-12-02
> >
> > I would like to thank the authors for the detailed response. Some of my concerns have been resolved.  However, the technical novelty of this paper is still incremental.
> > I will keep my original rating.

---

### Meta-Review · Area_Chair_NX17 · 2024-12-24

**Metareview:**

The paper proposes GenVP, a generative framework for solving and creating Raven’s Progressive Matrices (RPM) puzzles by leveraging contrastive hierarchical VAEs. The model introduces novel disentangled representations for rule-relevant and irrelevant factors, a Mixture of Experts (MoE) module for rule prediction, and a contrastive learning scheme to enhance generalization. Experiments show state-of-the-art (SOTA) performance in RPM-solving accuracy and out-of-distribution (OOD) generalization, as well as the capability to generate novel RPM puzzles.

Reviewer GBgR maintained a negative stance, primarily due to concerns about the limited generalizability of GenVP beyond RPM tasks, the reliance on annotated rules for training, and perceived limitations in interpretability during answer selection. Reviewer YBT2 criticized the incremental nature of the contributions, particularly the use of existing techniques like VAEs and contrastive loss. The authors provided detailed rebuttals and additional experiments addressing these concerns. For GBgR’s generalization concerns, they demonstrated GenVP's scalability to diverse datasets and problem settings, including rule-annotation-free training, which showed significant performance gains over prior models. Regarding the reliance on annotated rules, the authors emphasized consistency with prior methods and conducted unsupervised training experiments to showcase potential for broader applicability. For YBT2’s novelty concerns, the authors highlighted several distinct contributions, including disentangled latent spaces, high-level rule reasoning, and puzzle-wide contrastive loss, which differentiate GenVP from prior approaches.

Considering the detailed rebuttal, good empirical results, and improvements over prior methods, the AC recommends acceptance. The authors have addressed major concerns with additional evidence and clarified the unique contributions of their approach. However, future work should focus on demonstrating broader applicability and minimizing reliance on annotations to further solidify the framework's impact.

**Additional Comments On Reviewer Discussion:**

Please refer to the meta-review.

---

### Decision · Program_Chairs · 2025-01-22

Accept (Poster)